# FRACTAL GRAPH CONTRASTIVE LEARNING

## ABSTRACT

While Graph Contrastive Learning (GCL) has attracted considerable attention in the field of graph self-supervised learning, its performance heavily relies on data augmentations that are expected to generate semantically consistent positive pairs. Existing strategies typically resort to random perturbations or local structure preservation, yet lack explicit control over global structural consistency between augmented views. To address this limitation, we propose Fractal Graph Contrastive Learning (FractalGCL), a theory-driven framework introducing two key innovations: a renormalisation-based augmentation that generates structurally aligned positive views via box coverings; and a fractal-dimension-aware contrastive loss that aligns graph embeddings according to their fractal dimensions, equipping the method with a fallback mechanism guaranteeing a performance lower bound even on non-fractal graphs. While combining the two innovations markedly boosts graph-representation quality, it also adds non-trivial computational overhead. To mitigate the computational overhead of fractal dimension estimation, we derive a one-shot estimator by proving that the dimension discrepancy between original and renormalised graphs converges weakly to a centred Gaussian distribution. This theoretical insight enables a reduction in dimension computation cost by an order of magnitude, cutting overall training time by approximately 61%. The experiments show that FractalGCL not only delivers state-of-the-art results on standard benchmarks but also outperforms traditional and latest baselines on traffic networks by an average margin of about remarkably 4%. Codes are available at (https://anonymous.4open.science/r/FractalGCL-0511/).

## 1 INTRODUCTION

Graph contrastive learning (GCL) has emerged as a popular self-supervised paradigm for graph representation learning (Hu et al., 2020; Xia et al., 2022a; You et al., 2021; Ju et al., 2024; Liu et al., 2022a; Xu et al., 2018). By forcing models to discriminate positive pairs from negative pairs, it alleviates the knowledge scarcity problems (Wu et al., 2021; Xie et al., 2022; Chen et al., 2025; Shi et al., 2025), and it also serves as an effective pretext task for pre-training graph foundation models (Liu et al., 2023; Huang et al., 2024). As graphs possess non-Euclidean topology, researchers must tailor contrastive learning frameworks to graph-specific properties. Therefore, GCL has formed unique lines of research, which cover stages including augmenting graph data (Liu et al., 2022b; Rong et al., 2019; Sun et al., 2021), designing contrastive modes (Ju et al., 2023; Ren et al., 2021; Park et al., 2020), and optimizing contrastive objectives (Hjelm et al., 2018; Xia et al., 2022b; Zhang et al., 2022).

Among current studies, data augmentation remains a pivotal challenge in graph contrastive learning, as the quality of positive and negative sample pairs fundamentally determines the capacity of a graph model to extract meaningful knowledge and the quality of learned representations. While negative samples are typically generated by contrasting views from structurally distinct graphs or subgraphs (You et al., 2020), which ensures divergent distributional characteristics, the generation of semantically coherent positive samples remains a critical bottleneck. Specifically, existing approaches often rely on random perturbations (e.g., node/edge deletion, attribute masking) or fixed topological constraints (e.g., hierarchy preservation), which provide only incomplete guarantees for maintaining structural consistency. These methods lack an explicit mechanism to ensure global similarity between the original graph and its augmented views, leading to potential mismatches in semantic alignment. This gap naturally raises a fundamental question: *Can we design a principled graph-level criterion to enforce global structural consistency during positive sample generation?*

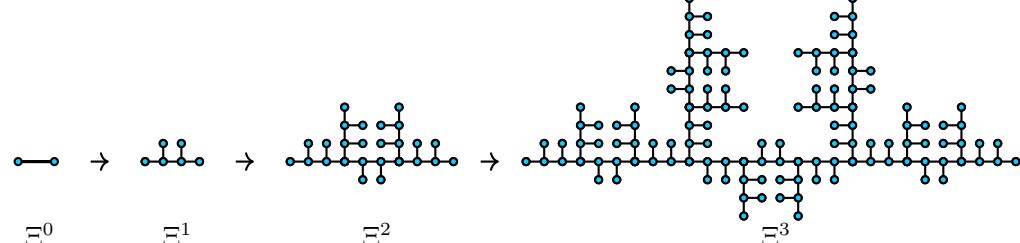

$$\Xi^0 \qquad \Xi^1 \qquad \Xi^2 \qquad \Xi^3$$

Figure 1: An example of evolving theoretical fractal graph (Neroli, 2024)

This critical question directs attention to a fundamental yet often overlooked global property of graphs—their inherent self-similarity and hierarchical complexity, which is mathematically formalised through the concept of **fractal**. Fractal geometry (Edgar & Edgar, 2008; Mandelbrot, 1983; 1989) is a field of mathematics that explores irregular shapes whose intricate detail persists across different scales, appearing in patterns such as snowflakes, coastlines, and branching trees. Fractal graphs are networks that possess fractal properties, effectively transplanting fractal concepts from Euclidean space onto graph structures (see Figure 1). Given the prevalence of fractal graphs in natural and society, their fractal properties likely play a significant yet under-explored role in improving graph representations via GCL.

To effectively utilize fractal properties, we propose a novel FractalGCL framework in this paper, improving the effectiveness of GCL. We start by introducing a novel augmentation strategy, **renormalisation**, to generate positive views which are structurally similar. Therefore, the generated views have the same box dimension, implying strong structural similarity. To ensure that the graph representations capture not only self-similar structures but also explicitly encode fractal-dimension information, we define a **fractal-dimension–aware contrastive loss** that steers the encoder to embed graphs in a way that respects their intrinsic fractal geometry. Empirically, the two components already outperform competing models, yet estimating the fractal dimension introduces additional computational overhead. Consequently, we cut the cost of box-dimension estimation with a theoretical result that approximates the dimension gap as a **Gaussian perturbation**, making FractalGCL practical and performant. Experiments were conducted on both standard graph classification benchmarks and real-world traffic networks, and the results confirm that FractalGCL surpasses prior methods on most individual benchmarks and attains the best average performance overall, underscoring its effectiveness in both theory and practice.

To sum up, our main contributions include:

- **Fractal Geometry Meets GCL.** To the best of our knowledge, we are among the first to inject a mathematically rigorous fractal viewpoint into graph representation learning and graph contrastive learning, revealing that a global and scale-free structure, which is often overlooked by prior GCL methods, demonstrates significant potential in learning high-quality graph representations and enhancing performance on downstream tasks.

- **Theory-Driven FractalGCL Architecture.** Guided by fractal geometry, we improve the existing GCL methods with a novel framework FractalGCL. It integrates renormalisation-based graph augmentations and a fractal-dimension–aware contrastive loss. Renormalisation contributes to generating better positive and negative pairs, while the novel loss further utilises fractal property to optimize graph embeddings.

- **From Theory to Implementation.** To further optimize Fractal GCL for practical implementation, we conducted a series of theoretical analyses to prove that the gap between the original and renormalised box dimensions converges weakly to a centred Gaussian measure, enabling a lightweight approach that markedly accelerates training. We design a principled fallback mechanism, ensuring that even on graphs without fractality our performance is no worse than classical GCL.

- **Notable Performance Gains.** We conducted thorough experiments on both standard graph classification benchmarks and social datasets, such as urban-traffic graphs, and the results prove the effectiveness of the proposed FractalGCL.

## 2  PRELIMINARY EXPERIMENTS

The foregoing discussion can be condensed into two working hypotheses: **(i)** fractal structures are widespread in real-world graphs, and **(ii)** such patterns reflect non-trivial global complexity that may influence representation learning.

We now present two preliminary studies to investigate these questions. *Preliminary experiment 1* measures how often strong fractality occurs in standard benchmarks, and *Preliminary experiment 2* evaluates whether explicitly using fractal information can boost downstream performance. Please refer to Appendix B for the detailed experimental setup and complete results.

**Preliminary experiment 1.** We assessed how well each graph in six graph classification benchmarks follows fractal (power-law) scaling by fitting a log–log box-counting linear regression and recording its **coefficient of determination** $R^2$: the closer $R^2$ is to 1, the more convincingly the graph is fractal.

Using the strict cutoff $R^2 \geq 0.90$, $81\%$ of the PROTEINS graphs, $92\%$ of the REDDIT-MULTI-5K graphs, and an impressive $99.8\%$ of the D&D graphs meet the criterion and similarly high ratios on the remaining datasets; see Figure 2. Hence, strongly fractal graphs are not rare outliers but a pervasive phenomenon across all six collections.

In fact, fractal behaviour (especially in larger graphs) is very common. Zakar-Polyák et al. (2023) In a corpus of 275 real-world networks, approximately 80% are classified as fractal. Ma et al. (2020) For Chinese urban street networks, about 67% exhibit fractal (power-law) behaviour; moreover, at the level of street connectivities, almost all cities display a fractal hierarchy. Laurienti et al. (2011) Examining 47 self-organising networks, 47/47 satisfy the fractal size–density scaling criterion.

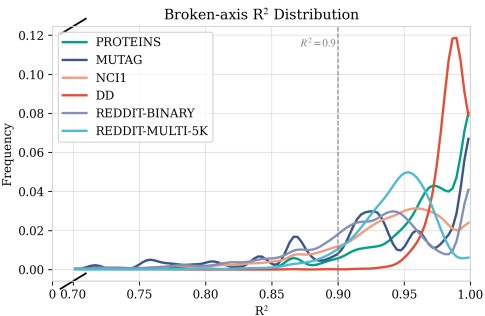

Figure 2: Prevalence of fractal structures

Figure 3: Accuracy gains from adding box dimension (significant when $p < 0.05$)

**Preliminary experiment 2.** We augmented a new feature **box dimension**, a type of fractal dimension introduced in Section 3.1, to the original features in each benchmark, and report the difference in classification accuracy before and after augmentation. Remarkably, four out of six benchmarks show *statistically significant* gains ($p < 0.05$), strongly indicating that fractal information captures unique topological features that previous models fail to capture.

## 3  FRACTALGCL: THEORY, METHODOLOGY AND IMPLEMENTATION

This section constructs FractalGCL — a novel framework grounded in fractal geometry that enables graph representations to capture global fractal structure and box dimension information at the graph level. Specifically, Section 3.1 revisits the essentials of fractal geometry; Sections 3.2–3.3 present our renormalisation-based augmentations and the accompanying dimension-aware contrastive loss, which form core components in FractalGCL framework; However, computing the fractal loss for each renormalised graph is extremely costly. Sections 3.4–3.5 address the resulting computational challenge by mathematical proof and statistical analysis and detail the practical implementation of FractalGCL. See Figure 4 for intuitive ideas.

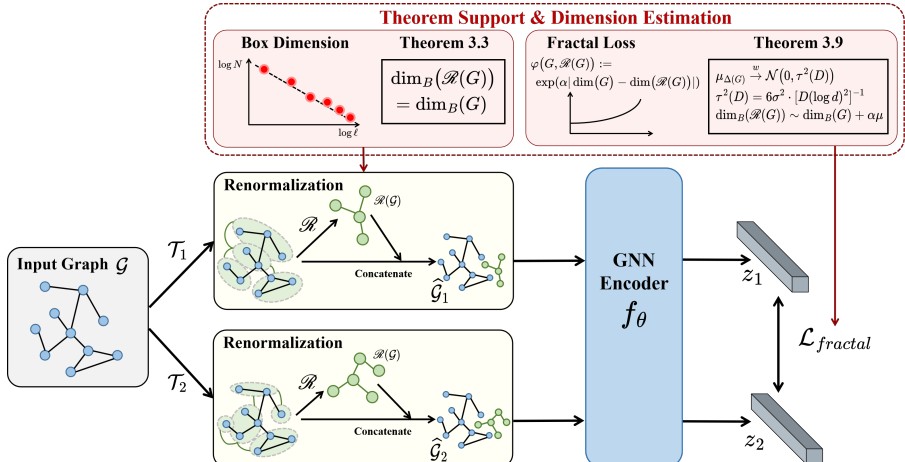

Figure 4: The Pipeline of FractalGCL

## 3.1 BACKGROUND OF FRACTAL GEOMETRY

In mathematics, defining a fractal graph is far from straightforward. However, to keep the discussion focused, we state the most practical definition of fractal dimension in plain terms here, leaving the full technical treatment to Appendix A.

**Definition 3.1.** *Let $G$ be an infinite graph equipped with the graph distance $d_G$. An $L$ box covering of $G$ is a collection of subgraphs $\{U_i\}_{i \in \mathcal{I}}$, indexed by $\mathcal{I}$, such that $\bigcup_{i \in \mathcal{I}} U_i = G$, and the diameter $\mathrm{diam}(U_i) \leq L$ for any $i \in \mathcal{I}$. Denote by $N_L(G)$ the minimum number of subgraphs required for an $L$ box covering of $G$. The **Minkowski dimension (or box dimension)** of $G$ is then defined as*

$$\dim_{\mathrm{B}}(G) := \lim_{L / \mathrm{diam}(G) \to 0} \frac{\log N_L(G)}{-\log\big(L / \mathrm{diam}(G)\big)},$$

*provided the limit exists. If $0 < \dim_{\mathrm{B}}(G) < \infty$, we say that $G$ exhibits a fractal property and call $G$ a fractal (Minkowski) graph. In network science, it can also be stated as $N_L(G) \sim \left(\frac{\mathrm{diam}(G)}{L}\right)^\beta$ for $L \ll \mathrm{diam}(G)$.*

To estimate the fractal dimension of a finite graph in practice, we design the algorithm estimating the box dimension. It forms the basis on which FractalGCL is built. Find full details on the box dimension algorithm in Appendix A.

## 3.2 NEW AUGMENTATION: GRAPH RENORMALISATION

In this section, we first formally introduce the renormalised graph $\mathscr{R}(G)$ and highlight its distinctive value when used as a novel augmentation in contrastive learning. We then present theoretical results showing that the renormalisation procedure preserves the fractal dimension of a graph, thereby providing a solid analytical foundation for our approach.

**Renormalisation Graph.**

In contrastive learning, constructing an "augmented graph" that retains structural similarity while preserving appropriate differences from the original is crucial. Here, we introduce the concept of the *renormalisation graph*, whose core idea has appeared in various literature (e.g., in multi-scale network analysis and abstractions of complex networks (Song et al., 2005)), but whose application to augmentations for contrastive learning is relatively novel.

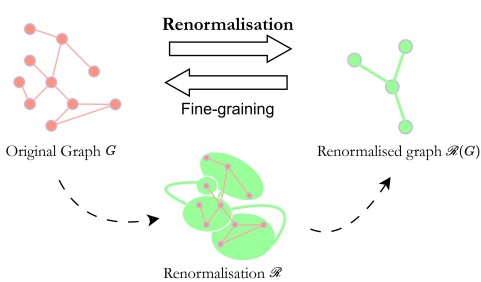

Figure 5: Graph Renormalisation

**Definition 3.2.** *Let $G$ be a given graph, and let $\{U_i\}_{i\in\mathcal{I}}$ be an $L$ box-covering of $G$. We construct the renormalised graph $\mathscr{R}(G)$ as follows: (i) Collapse each covering set $U_i$ into a single* supervertex $v_i$. *(ii) If there exists at least one edge in $G$ connecting a vertex in $U_i$ to a vertex in $U_j$ (with $i \neq j$), then place a* superedge *between $v_i$ and $v_j$ in $\mathscr{R}(G)$. The resulting graph $\mathscr{R}(G)$ is called the* renormalised graph *of $G$ at the given scale $L$. $\mathscr{R}(G)$ is equipped with the unweighted shortest-path metric induced by its adjacency matrix.*

From an intuitive standpoint, the construction of $\mathscr{R}(G)$ disregards certain fine-grained local structures while highlighting the global characteristics of the original graph in a more compact form. Because renormalisation at different scales can accentuate multi-scale self-similarity, having both $G$ and $\mathscr{R}(G)$ simultaneously in contrastive learning allows the model to "perceive" macro-level structural resemblance, thereby facilitating a more effective capture of the essential features of a fractal network. Full algorithmic details are deferred to Appendix A.

**Theorem 3.3.** *For any Minkowski (box-dimensional) infinite graph $G$, mathematically,*

$$\dim_{\mathrm{B}}\big(\mathscr{R}(G)\big) = \dim_{\mathrm{B}}(G).$$

*Proof.* See Theorem A.2 in Appendix. $\qquad\qquad\qquad\qquad\qquad\qquad\qquad\qquad\qquad\square$

Note that for any graph $G$ of infinite diameter, its renormalisation $\mathscr{R}(G)$ necessarily retains infinite diameter; hence the Minkowski dimension remains well-defined. Theorem 3.3 formally states that the box dimension is invariant under renormalisation, ensuring that $\mathscr{R}(G)$ and $G$ share the same intrinsic fractal complexity.

In experimental practice, we construct the augmentation view as the disjoint union $G \sqcup \mathscr{R}(G)$. Because $\mathscr{R}(G)$ preserves both the fractal dimension and the self-similar structure of $G$, it serves as a scaled-down fractal module drawn from the same generative process. Appending this module to $G$ enlarges the global pattern while introducing controlled local variation, producing an augmented graph that is recognizably similar yet still distinguishable from the original. See Figure 4.

### 3.3 NOVEL LOSS: FRACTAL-DIMENSION BASED

While renormalisation already captures the fractal structure, in this section we introduce a contrastive loss with fractal dimension. Together, these components yield a graph representation learning framework that embeds each graph's fractal characteristics.

**Mapping from Graph $G$ to Representation z.** we apply $\mathscr{R}$ to obtain the augmented graph $\mathscr{R}(G_n)$, where $G_n$ is the n-th original graph in a mini-batch. We then use a GNN-based encoder $f_\theta(\cdot)$ and a readout function $\mathrm{Readout}(\cdot)$ to produce a graph-level embedding, and finally apply a projection head $g_\phi(\cdot)$ to map it into the contrastive space: $\mathbf{z}_n := g_\phi\Big(\mathrm{Readout}\big(f_\theta(\mathscr{R}(G_n))\big)\Big)$.

**Contrastive Loss with Fractal Weight.** We define the fractal dimension discrepancy weight between $G_n$ and its augmented version $\mathscr{R}(G_n)$ as

$$\varphi\big(G_n, \mathscr{R}(G_n)\big) := \exp\Big(\alpha\big|\dim_{\mathrm{B}}(G_n) - \dim_{\mathrm{B}}\big(\mathscr{R}(G_n)\big)\big|\Big),$$

where $\dim_{\mathrm{B}}(\cdot)$ denotes the (estimated) Minkowski dimension of a graph, and $\alpha \geq 0$ is a scaling factor.

Assume we have $N$ original graphs $\{G_n\}_{n=1}^N$ in a minibatch. Each $G_n$ is augmented to produce $(G_n, \mathscr{R}(G_n))$, yielding representations $\mathbf{z}_n$ and $\mathbf{z}_n^{(\mathscr{R})}$, respectively. We treat $(\mathbf{z}_n, \mathbf{z}_n^{(\mathscr{R})})$ as a *positive pair* in the spirit of contrastive learning, while representations from other graphs in the batch serve as *negative* examples. An InfoNCE-like loss with the fractal dimension weight is given by:

$$\ell_{\mathrm{fractal}}(n) := -\log \frac{\exp\Big(\mathrm{sim}\big(\mathbf{z}_n, \mathbf{z}_n^{(\mathscr{R})}\big)/\tau\Big) \cdot \varphi\big(G_n, \mathscr{R}(G_n)\big)}{\sum_{n'=1, n'\neq n}^N \exp\Big(\mathrm{sim}\big(\mathbf{z}_n, \mathbf{z}_{n'}\big)/\tau\Big) \cdot \varphi\big(\mathscr{R}(G_n), \mathscr{R}'(G_{n'})\big)}.$$

We average over all $n$ to obtain the overall *fractal contrastive loss*: $\mathcal{L}_{\mathrm{fractal}} := \frac{1}{N}\sum_{n=1}^N \ell_{\mathrm{fractal}}(n)$.

**Lemma 3.4.** *Denote the similarity by $s(G) := \text{sim}(\mathbf{z}_G, \mathbf{z}_{\mathscr{R}(G)})$ and $\Delta(G) := \dim_{\mathrm{B}}(G) - \dim_{\mathrm{B}} \mathscr{R}(G)$. Keeping all other batch terms fixed, the fractal-weighted InfoNCE loss $\ell_{\text{fractal}}$ satisfies*

$$\left| \partial \ell_{\text{fractal}} / \partial s(G) \right| = w \left| \partial \ell_{\text{InfoNCE}} / \partial s(G) \right|, \; w = \exp(\alpha \Delta(G)) \text{ increases strictly with } \Delta(G).$$

*Proof.* Straightforward by partial differentiation. □

Intuitively, a larger fractal-dimension gap $\Delta(G)$ amplifies the positive-pair gradient, forcing the model to pull the two views closer, while $\Delta(G) = 0$ reduces to the ordinary InfoNCE case. The lemma therefore formalises how the weight $\exp(\alpha \Delta(G))$ adaptively injects fractal similarity into the optimisation dynamics.

**Proposition 3.5** (Dimension–Dominated Ranking Consistency). *If $\Delta(H_1) < \Delta(H_2)$ and $s(H_1) - s(H_2) \leq \tau\alpha(\Delta(H_2) - \Delta(H_1))$, then the fractal-weighted InfoNCE losses satisfy*

$$\ell_{\text{fractal}}(G, H_2) < \ell_{\text{fractal}}(G, H_1).$$

*Proof.* See Appendix A.3. □

Proposition 3.5 shows that when two candidates have nearly identical embedding similarities, the fractal-weighted InfoNCE loss favours the one whose fractal dimension is closer to that of the anchor graph, ensuring that fractal characteristics dominate the loss's discriminative behaviour.

### 3.4 COMPUTATIONAL DILEMMA AND ITS SOLUTION

In practice, the renormalisation augmentation combined with the fractal-dimension loss already yields strong downstream performance, but computing that loss for every renormalised graph is computationally expensive. This section analyzes the bottleneck and presents an efficient remedy.

To conclude, in practice we encounter the following dilemma:

**(I)** Simply imposing equal dimensions before and after renormalisation by Theorem 3.3, as guaranteed asymptotically by the theorem, overlooks the discrepancies that arise in finite graphs.

**(II)** Conversely, computing the fractal dimension for every augmented graph is prohibitively complex. The following Proposition 3.6 indicates that the per-augmentation estimation is unrealistic.

**Proposition 3.6** (Fractal complexity on sparse graphs). *For the greedy box-covering procedure in Algorithm 1, the worst-case running time $T(V)$ obeys $\Omega(V^2) \leq T(V) \leq O(V^3)$.*

*Proof.* See Appendix A.4. □

**A Theoretical Solution to the Dimension Dilemma.** We avoid the heavy cost of recomputing $\dim_{\mathrm{B}}(\mathscr{R}(G))$ at every augmentation step by modelling the finite-size deviation $\Delta(G) := \dim_{\mathrm{B}}(G) - \dim_{\mathrm{B}}(\mathscr{R}(G))$ as a random perturbation whose variance vanishes as the graph grows. The argument proceeds in three succinct steps.

**Step 1. Finite-diameter error magnitude.** Denote the diameter of a graph $G$ by $\text{diam}(G)$. Write $\hat{m}_G$ for the OLS slope used to estimate $\dim_{\mathrm{B}}(G)$ and $\sigma^2$ for the log–residual variance.

**Lemma 3.7** (Standard error vs. diameter). $\text{SE}(\hat{m}_G) \sim 2\sqrt{6}\sigma\left[\sqrt{\text{diam}(G)} \log \text{diam}(G)\right]^{-1}$.

*Proof.* See Appendix A.5 for details. □

Lemma 3.7 quantifies how rapidly the slope uncertainty shrinks: the error decays as $1/(\sqrt{\text{diam}(G)} \log \text{diam}(G))$.

**Step 2. Asymptotic distribution of the slope.**

**Lemma 3.8.** *Under Assumptions (A1)–(A4),*

$$\sqrt{\operatorname{diam}(G)}(\hat{m}_G - m_G) \xrightarrow{\mathscr{D}} \mathcal{N}(0, \sigma^2), \qquad \sqrt{\operatorname{diam}(G)}(\hat{m}_{\mathscr{R}} - m_{\mathscr{R}}) \to \mathcal{N}(0, \sigma^2).$$

*Proof.* Immediate from the classical OLS central-limit theorem. □

Lemma 3.8 states that, once rescaled by $\sqrt{\operatorname{diam}(G)}$, the slope estimator for either graph becomes asymptotically Gaussian with a diameter-independent variance $\sigma^2$. Hence any finite-size fluctuation of the estimated dimension is fully captured by a normal term whose magnitude is controlled only by $\operatorname{diam}(D)$.

**Step 3. Weak convergence of the dimension gap.**

**Theorem 3.9.** *Let $\mu_G$ be the probability measure induced by the random variable $\Delta(G)$ on a graph $G$. Under Hypothesis A.6 and with the notation of Lemmas 3.7 and 3.8, we have*

$$\mu_G \xrightarrow{w} \mathcal{N}\big(0, \kappa^2(\operatorname{diam}(G))\big), \qquad \kappa^2(\operatorname{diam}(G)) := 6\sigma^2[\operatorname{diam}(G)(\log \operatorname{diam}(G))^2]^{-1},$$

*as $\operatorname{diam}(G) \to \infty$. In particular, $\kappa^2(\operatorname{diam}(G)) \to 0$, so the limiting distribution degenerates to the Dirac measure $\delta_0$; i.e. $\Delta(G) \to 0$ in probability.*

*Proof.* See Appendix A.7. □

**Summary.** Accordingly, we estimate the renormalised graph's dimension for *every* $G$ by adding a zero-mean Gaussian perturbation with this scale, rather than rerunning the full box-covering procedure. This scale-adaptive stochastic perturbation preserves fractal information while replacing the prohibitive deterministic computation with an analytically grounded, lightweight approximation.

## 3.5 PRACTICAL IMPLEMENTATION

In this section, we integrate the newly developed methods and theory to implement the FractalGCL.

**Loss approximation.** For a minibatch $\{G_1, \ldots, G_N\}$ we draw independent perturbations

$$\mu_n \sim \mathcal{N}\big(0, \kappa^2(D_n)\big), \quad \nu_{nk} \sim \mathcal{N}\big(|\dim_{\mathrm{B}}(G_n) - \dim_{\mathrm{B}}(G_k)|, \kappa^2(\operatorname{diam}(G_n)) + \kappa^2(\operatorname{diam}(G_k))\big),$$

where $\hat{\sigma} \approx 0.1$ is the pilot-estimated residual scale. The fractal loss then reads $\ell_n^{\mathrm{fractal}} = -\log \dfrac{\exp\big(\operatorname{sim}(\mathbf{z}_n, \mathbf{z}_n^{(\mathscr{R})})/\kappa + \alpha \mu_n\big)}{\displaystyle\sum_{k \neq n} \exp\big(\operatorname{sim}(\mathbf{z}_n, \mathbf{z}_k^{(\mathscr{R})})/\kappa + \alpha \nu_{nk}\big)}$.

**Implementation details.** During training we first compute (or cache) each graph diameter $\operatorname{diam}(G_i)$, then form the similarity matrix $\mathbf{S} = [\operatorname{sim}(\mathbf{z}_i, \mathbf{z}_j^{(\mathscr{R})})]$ and augment it with a Gaussian matrix whose entrywise statistics obey the diameter–controlled variance above:

$$\mathbf{S}^* = \mathbf{S} + \alpha \mathbf{G}, \qquad \mathbf{G}_{ij} \sim \begin{cases} \mathcal{N}\big(0, \kappa^2(\operatorname{diam}(G_i)_i)\big), & i = j, \\ \mathcal{N}\big(|\dim_{\mathrm{B}}(G_i) - \dim_{\mathrm{B}}(G_j)|, \ \kappa^2(\operatorname{diam}(G_i)) + \kappa^2(\operatorname{diam}(G_j))\big), & i \neq j. \end{cases}$$

An annealing schedule on $\hat{\sigma}$ (or directly on $\alpha$) keeps the injected noise large in early epochs and negligible later. Softmax over $\mathbf{S}^*$ yields the final fractal-weighted contrastive loss, adding only at almost the $\mathcal{O}(N^2)$ cost of sampling $\mathbf{G}$ to each batch.

## 3.6 SAFE FALLBACK UNDER WEAK FRACTALITY OR SMALL DIAMETERS.

We employ a two-stage gate: if the graph diameter $\operatorname{diam}(G) \leq 9$ (for which box-dimension estimation is not meaningful) or the fractality is insufficient $R^2 < \theta$ with default $\theta = 0.9$, we disable the renormalised view and the fractal weighting by setting $\alpha = 0$, and retain only standard GCL local augmentations (e.g., node dropping). In this case the positive-pair weight is $\exp(\alpha \Delta(G)) = 1$, so the objective and its gradients reduce exactly to InfoNCE, and the method strictly degenerates to the GCL baseline in these regimes. Consequently, regardless of dataset fractality, the worst-case performance is at least **as good as** the corresponding GCL baseline. See Section 4.7 and Appendix C for parameter analysis.

## 4 EXPERIMENTS

### 4.1 SETUP

We validate FractalGCL on unsupervised representation learning tasks using six widely-adopted datasets from TUDataset (Morris et al., 2020): NCI1, MUTAG, PROTEINS, D&D, REDDIT-BINARY(REDDIT-B), and REDDIT-MULTI-5K(REDDIT-M5K). We adopt a 2-layer GIN as the encoder, and a sum pooling is used as the readout function; renormalisation adopts greedy box-covering with radius 1, dimension weight $\alpha = 0.1$, and temperature $\tau = 0.4$. Models are first trained with Adam on the unlabeled data only. After that, a non-linear SVM classifier is used to evaluate the graph representations. Accuracy is reported under 10-fold cross-validation. The experiments are repeated 5 times to report the mean and standard deviation. We conduct our experiments on an Ubuntu machine with one 40GB NVIDIA A100 GPU.

### 4.2 MAIN RESULTS

Table 1: Classification accuracy on benchmark datasets (10-fold CV).

| Model | NCI1 | MUTAG | PROTEINS | D&D | REDDIT-B | REDDIT-M5K | AVG. |
|---|---|---|---|---|---|---|---|
| GAE (Kipf & Welling, 2016) | 74.36 ± 0.24 | 72.87 ± 6.84 | 70.51 ± 0.17 | 74.54 ± 0.68 | 87.69 ± 0.40 | 33.58 ± 0.13 | 68.93 ± 1.41 |
| graph2vec (Narayanan et al., 2017) | 73.22 ± 1.81 | 83.15 ± 9.25 | 73.30 ± 2.05 | 70.32 ± 2.32 | 75.48 ± 1.03 | 47.86 ± 0.26 | 70.56 ± 2.79 |
| DGI (Velickovic et al., 2019) | 74.86 ± 0.26 | 66.49 ± 2.28 | 72.27 ± 0.40 | 75.78 ± 0.34 | 88.66 ± 0.95 | 53.61 ± 0.31 | 71.95 ± 0.76 |
| InfoGraph (Sun et al., 2019) | 76.20 ± 1.06 | 89.01 ± 1.13 | 74.44 ± 0.31 | 72.85 ± 1.78 | 82.50 ± 1.42 | 53.46 ± 1.03 | 74.74 ± 1.12 |
| GraphCL (You et al., 2020) | 77.87 ± 0.41 | 86.80 ± 1.34 | 74.39 ± 0.45 | 78.62 ± 0.40 | 89.53 ± 0.84 | 55.99 ± 0.28 | 77.20 ± 0.62 |
| ContextPred (Hu et al., 2020) | 73.00 ± 0.30 | 71.75 ± 7.34 | 70.23 ± 0.63 | 74.66 ± 0.51 | 84.76 ± 0.52 | 51.23 ± 0.84 | 70.94 ± 1.69 |
| JOAO (You et al., 2021) | 78.07 ± 0.47 | 87.35 ± 1.02 | 74.55 ± 0.41 | 77.32 ± 0.54 | 85.29 ± 1.35 | 55.74 ± 0.63 | 76.39 ± 0.74 |
| JOAOv2 (You et al., 2021) | 78.36 ± 0.53 | 87.67 ± 0.79 | 74.07 ± 1.10 | 77.40 ± 1.15 | 86.42 ± 1.45 | 56.03 ± 0.27 | 76.66 ± 0.88 |
| SimGRACE (Xia et al., 2022a) | 79.12 ± 0.44 | 89.01 ± 1.31 | 74.03 ± 0.09 | 77.44 ± 1.11 | 89.51 ± 0.89 | 55.91 ± 0.34 | 77.50 ± 0.70 |
| RGCL (Li et al., 2022) | 78.14 ± 1.08 | 87.66 ± 1.01 | 75.03 ± 0.43 | 78.86 ± 0.48 | 90.34 ± 0.58 | 56.38 ± 0.40 | 77.74 ± 0.66 |
| DRGCL (Ji et al., 2024) | 78.70 ± 0.40 | 89.50 ± 0.60 | 75.20 ± 0.60 | 78.40 ± 0.70 | **90.80 ± 0.30** | 56.30 ± 0.20 | 78.15 ± 0.47 |
| GradGCL (Li et al., 2024) | 79.72 ± 0.53 | 88.46 ± 0.98 | 74.89 ± 0.39 | 78.95 ± 0.47 | 90.45 ± 1.06 | 56.20 ± 0.31 | 78.11 ± 0.62 |
| FractalGCL | **80.50 ± 0.16** | **91.71 ± 0.23** | **75.85 ± 0.40** | **81.71 ± 0.57** | 90.41 ± 0.72 | **57.29 ± 0.59** | **79.58 ± 0.45** |

Table 1 reports the accuracy of the downstream graph classification task on six benchmark datasets. FractalGCL achieves the highest average score (**79.58**%), outperforming the strongest baseline (GradGCL, 78.15%) by **1.43 pp**. It ranks first on five of the six datasets—**NCI1**, **MUTAG**, **PRO-TEINS**, **D&D**, and **REDDIT-MULTI-5K**. The most strongly fractal benchmark D&D exhibits a big margin (**2.76pp**), which is consistent with our hypothesis that fractal-aware augmentations and loss provide greater benefit when the underlying graphs display pronounced self-similarity. These results confirm that injecting the fractal structure into a graph contrastive learning not only matches but often exceeds the performance of carefully tuned augmentation-based methods, while retaining the same encoder capacity and training budget.

### 4.3 EVALUATION ON PRACTICAL SCENARIOS

To assess the real-world applicability of FractalGCL, we followed network collection way in (Zhai et al., 2025) to construct urban road graphs for Chicago, New York and San Francisco. We then randomly sampled square sub-graphs from each complete road network. The downstream task predicts the traffic-accident severity of each area, following (Zhao et al., 2024). Full experimental details and results are provided in Appendix B.

Table 2: Classification accuracy on traffic tasks.

| Task | City | DGI | InfoGraph | GCL | JOAO | SimGRACE | DRGCL | GradGCL | **FractalGCL** |
|---|---|---|---|---|---|---|---|---|---|
| total_accidents_high | Chicago | 54.91±10.75 | 56.54±11.85 | 63.12±13.36 | 55.86±12.21 | 62.75±13.50 | 63.49±16.04 | 63.55±18.73 | **64.60±13.32** |
| | SF | 76.45±14.47 | 78.74±13.60 | 80.06±13.32 | 79.75±13.61 | 80.40±13.47 | 80.81±10.69 | 80.87±9.62 | **80.89±12.92** |
| | NY | 51.51±7.62 | 51.85±8.82 | 55.83±12.38 | 51.10±11.01 | 52.27±12.06 | 58.06±14.82 | 57.33±18.01 | **68.39±13.84** |
| accident_volume_level | Chicago | 43.09±11.96 | 42.77±12.15 | 46.90±13.71 | 43.15±12.55 | 46.26±13.34 | 47.23±14.11 | 47.70±16.21 | **48.83±13.68** |
| | SF | 55.85±11.48 | 58.07±10.67 | 58.31±10.86 | 58.40±10.61 | **58.43±11.10** | 58.31±9.84 | 58.88±8.86 | 58.10±11.37 |
| | NY | 37.14±10.59 | 35.72±10.21 | 39.22±13.34 | 35.86±12.14 | 36.01±13.09 | 40.52±18.18 | 39.28±19.86 | **50.17±14.79** |
| risk_level | Chicago | 34.19±7.29 | 34.02±7.68 | 40.49±12.05 | 33.90±9.30 | 37.74±11.12 | 40.60±19.75 | 40.82±15.30 | **43.95±12.63** |
| | SF | 39.47±11.15 | 40.98±10.84 | 41.26±11.59 | 41.76±12.48 | 41.41±11.76 | 41.02±16.33 | 41.89±18.90 | **42.34±12.20** |
| | NY | 41.46±11.21 | 42.06±11.21 | 47.30±13.63 | 41.40±12.38 | 42.80±13.05 | 49.84±9.38 | 48.02±13.53 | **57.24±14.23** |
| Average | – | 48.23±10.92 | 48.97±10.91 | 52.50±12.73 | 49.02±11.87 | 50.90±12.53 | 53.32±14.77 | 53.15±15.91 | **57.17±13.26** |

Table 2 summarizes three downstream tasks across three cities, yielding nine classification settings in total. FractalGCL attains the highest accuracy in eight of the nine settings and lifts the overall

average to **57.17%**, an impressive **3.85%** lead over the next-best model, DRGCL (53.32%). We attribute this gain to the strongly fractal nature of urban networks, which makes a fractal-aware approach especially effective; further details appear in Appendix B and publicly available code.

## 4.4 VALIDATION OF THEOREM 3.9

We aim to test that the change in the Minkowski dimension after one-step renormalisation $\Delta(G)$ is Gaussian in distribution. On the TUDataset D&D benchmark, we find $\mathrm{mean}\Delta(G) = -0.1084$ and $\mathrm{std} = 0.1058$ with $n = 1178$; The regression $D' = \alpha + \beta D$ yields $\alpha = 0.0512$, $\beta = 0.9711$ and $R^2 = 0.87$, and $\mathrm{corr}(D, D' - D) = 0.0214$ ($p = 0.68$). Hence, the experimental evidence supports Theorem 3.9.

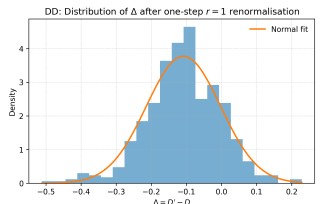

Figure 6: Gaussion validation

## 4.5 ABLATION STUDY

Table 3: **(a) Ablation study**

| Method | Components | | MUTAG | |
| --- | --- | --- | --- | --- |
| | Ren. | Frac. Loss | Acc. | Time (s) |
| FractalGCL | ✓ | ✓ | 91.71 | 486.81 |
| w/o. Graph Concat | ✓ | ✓ | 90.41 | 321.87 |
| w/o. Renormalisation | ✗ | ✓ | 88.46 | 33.93 |
| w/o. Fractal Loss | ✓ | ✗ | 88.09 | 423.97 |
| w/. Exact Dimension | ✓ | ✓ | **91.93** | 1249.74 |

Table 4: **(b) Variant accuracy**

| Variant | D&D | MUTAG |
| --- | --- | --- |
| FractalGCL | **81.71** | **91.71** |
| + random radius | 80.05 | 88.73 |
| + $R^2$ prob. | 81.12 | 88.83 |
| $- R^2$ threshold | 80.94 | 88.33 |

Table 3 lists MUTAG accuracy and pre-training time as we remove FractalGCL's three key components—graph concatenation, renormalisation, and the fractal-dimension loss—one at a time. Dropping any single component lowers accuracy by about 1.3–3.6 pp, confirming that each part is essential. In terms of efficiency, our Gaussian surrogate for box-dimension estimation trims training time from 1249.74 s (w/. Exact Dimension) to 486.81 s, nearly a $2.56 \times$ speed-up—that is, roughly a 61% reduction in compute.

## 4.6 VARIANT EXPERIMENTS

Table 4 compares three ways of altering the renormalisation rule. Introducing a random radius or discarding the fractality filter both weaken the structural match between views and lower accuracy, while using $R^2$ merely as a soft sampling probability yields a middle-ground result. These variants confirm that a fixed small radius combined with an explicit $R^2$ threshold offers the best balance between view diversity and global consistency.

## 4.7 PARAMETER ANALYSIS

Figure 7 shows that FractalGCL is robust to reasonable hyper-parameter changes: accuracy varies within about one percentage point across all tested settings. A moderate fractality filter ($R^2 \approx 0.9$) and a small dimension weight ($\alpha \leq 0.3$) already captures most of the gain, while larger

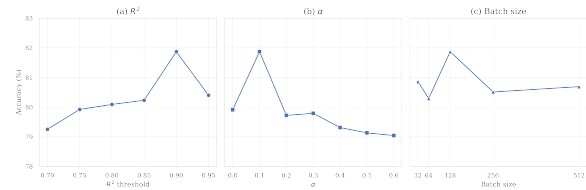

Figure 7: Hyper-parameter sensitivity on D&D

penalties or very loose filters begin to erode performance. Batch size has little impact, confirming that the method scales smoothly without delicate tuning of training throughput. Also See Appendix C.

## 5 RELATED WORKS AND CONCLUSIONS

See Appendix D for Related Works and Appendix G for Conclusions.

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

# APPENDIX

## A    PROOFS

A rigorous mathematical approach to define and analyse "fractal graphs" relies on viewing a graph as a metric space and studying its *scaling limit* in the sense of Gromov-Hausdorff topology. Concretely, in this section, let $G = (V, E)$ be a simple, connected infinite graph with its shortest-path metric. A sequence of such graphs $(G^n)$ is said to converge to a limiting metric space $G^\infty$ if

$$\lim_{n \to \infty} d_{\text{GH}}(G^n, G^\infty) = 0,$$

where $d_{\mathrm{GH}}$ is the Gromov–Hausdorff distance. If the limit $G^\infty$ exhibits fractal behaviour, then the original sequence $(G^n)$ is often viewed to possess fractality in a limiting sense.

Although this framework is theoretically well-founded and widely studied in the context of metric geometry and fractal analysis, it typically introduces extensive technical details. In real-world problems involving large-scale networks (e.g., deep neural architectures, biological networks, or social graphs), a full treatment of Gromov–Hausdorff convergence can be unnecessarily complex. Consequently, the present work uses the notion of "infinite graphs" and "fractal-like structures" primarily as an intuitive and useful abstraction of multi-scale patterns, rather than relying on a strict Gromov-Hausdorff scaling limit argument. Readers interested in the detailed mathematical background are referred to (Falconer, 2013; Gromov et al., 1999; Neroli, 2024) for further discussion.

**Definition A.1** (Definition 3.1). *Let $G$ be an (infinite) graph equipped with a graph distance $d_G$. An $L$ box-covering of $G$ is a collection of subgraphs $\{U_i\}_{i \in \mathcal{I}}$ where $\mathcal{I}$ is an index set, such that:*

$$\bigcup_{i \in \mathcal{I}} U_i = G \quad and \quad \mathrm{diam}(U_i) \leq L \quad \forall i \in \mathcal{I}.$$

*Here, $\mathrm{diam}(U_i)$ refers to the diameter of $U_i$ regarding metric $d_G$. We denote by $N_L(G)$ the minimum number of subgraphs needed for an $L$ box-covering of $G$. Then the* Minkowski dimension *(also called the* box dimension*) of $G$ is given by*

$$\dim_{\mathrm{B}}(G) := \lim_{L/\mathrm{diam}(G) \to 0} \frac{\log N_L(G)}{-\log \frac{L}{\mathrm{diam}(G)}},$$

*provided this limit exists.*

*If $\dim_{\mathrm{B}}(G)$ is both finite and strictly positive, we say that $G$ possesses a fractal property, and we refer to $G$ as a* Minkowski graph.

**Theorem A.2** (Theorem 3.3).
$$\dim_{\mathrm{B}}\big(\mathscr{R}(G)\big) = \dim_{\mathrm{B}}(G).$$

*Proof.* Denote by $N_L(G)$ the minimum number of $L$-box-covering sets of $G$, and let $N_L\big(\mathscr{R}(G)\big)$ be the analogous quantity for the renormalised graph.

Any $L$-covering of $G$ naturally induces an $L$-covering of $\mathscr{R}(G)$. Indeed, since each "supervertex" in $\mathscr{R}(G)$ corresponds to one of the $L$-boxes in $G$, you can treat each box as if it were "collapsed" into a single node. Therefore,
$$N_L\big(\mathscr{R}(G)\big) \leq N_L(G).$$

Conversely, given an $L$-covering of $\mathscr{R}(G)$, one can "expand" each supervertex $v_i$ back to the corresponding box $U_i$ in $G$. Since edges between two supervertices in $\mathscr{R}(G)$ indicate there was a connection between the respective $U_i$ and $U_j$ in $G$, the covering in $\mathscr{R}(G)$ lifts to an $L'$-covering of $G$ (where $L'$ is of the same order as $L$, up to a possible constant factor). Hence we obtain a bound of the form
$$N_L(G) \leq c N_L\big(\mathscr{R}(G)\big),$$
for some absolute constant $c$.

Combining these bounds yields $c N_L(G) \leq N_L\big(\mathscr{R}(G)\big) \leq N_L(G)$, where $c$ is a positive constant independent of $L$. Finally,

$$\dim_{\mathrm{B}}(G) = \lim_{L/\mathrm{diam}(G) \to 0} \frac{\log\big(c N_L(G)\big)}{-\log \frac{L}{\mathrm{diam}(G)}}$$

$$\leq \lim_{L/\mathrm{diam}(G) \to 0} \frac{\log\big(N_L(G) + \log c\big)}{-\log \frac{L}{\mathrm{diam}(G)}}$$

$$\leq \dim_{\mathrm{B}}(\mathscr{R}(G)) \leq \dim_{\mathrm{B}}(G).$$

$\square$

**Algorithm 1** Algorithm of computing box dimension

---

**Require:** Graph $\mathcal{G}$ with node set $\mathcal{V}$ and diameter $d$.
**Ensure:** Fractality metric $R^2$ and box dimension $\dim_{\mathrm{B}}(\mathcal{G})$.

1: **if** $d \leq 9$ **then**                                         ▷ too small for fractal analysis
2:     $R^2 \leftarrow 0$.
3:     $\dim_{\mathrm{B}}(\mathcal{G}) \leftarrow 0$.
4: **else**
5:     $L_{\max} \leftarrow \lfloor d/2 \rfloor$.
6:     $Array \leftarrow \varnothing$.
7:     **for** $l \leftarrow 1$ to $L_{\max}$ **do**
8:         $r \leftarrow \lfloor l/2 \rfloor$.
9:         $\mathcal{V}_{\mathrm{remain}} \leftarrow \mathcal{V}$.
10:         $N_B(l) \leftarrow 0$.
11:         **if** $l$ is even **then**
12:             **while** $\mathcal{V}_{\mathrm{remain}} \neq \varnothing$ **do**
13:                 $v \leftarrow \arg\max_{v \in \mathcal{V}_{\mathrm{remain}}} |B(v,r)|$.         ▷ $B(v,r) = \{i \in \mathcal{V}_{\mathrm{remain}} \mid d_{\mathcal{G}}(i,v) \leq r\}$
14:                 $\mathcal{V}_{\mathrm{remain}} \leftarrow \mathcal{V}_{\mathrm{remain}} \setminus B(v,r)$.
15:                 $N_B(l) \leftarrow N_B(l) + 1$.
16:             **end while**
17:         **else**                                         ▷ $l$ is odd
18:             **if** $\exists v, w \in \mathcal{V}_{\mathrm{remain}}$ with $d_{\mathcal{G}}(v,w) = 1$ **then**
19:                 $(v,w) \leftarrow \arg\max_{v,w \in \mathcal{V}_{\mathrm{remain}}} |B(v,r) \cup B(w,r)|$.
20:                 $\mathcal{V}_{\mathrm{remain}} \leftarrow \mathcal{V}_{\mathrm{remain}} \setminus \big(B(v,r) \cup B(w,r)\big)$.
21:                 $N_B(l) \leftarrow N_B(l) + 1$.
22:             **else**                                         ▷ fallback to the even-$l$ single-centre case
23:                 $v \leftarrow \arg\max_{v \in \mathcal{V}_{\mathrm{remain}}} |B(v,r)|$.
24:                 $\mathcal{V}_{\mathrm{remain}} \leftarrow \mathcal{V}_{\mathrm{remain}} \setminus B(v,r)$.
25:                 $N_B(l) \leftarrow N_B(l) + 1$.
26:             **end if**
27:         **end if**
28:         $Array \leftarrow Array \cup \{(\log l,\ \log N_B(l))\}$.
29:     **end for**
30:     Fit $y = mx + b$ to $Array$ by least squares and compute $R^2$.
31:     $\dim_{\mathrm{B}}(\mathcal{G}) \leftarrow -m$.
32: **end if**
33: **return** $R^2$, $\dim_{\mathrm{B}}(\mathcal{G})$.

---

**Proposition A.3** (Dimension–Dominated Ranking Consistency). *If $\Delta(H_1) < \Delta(H_2)$ and $s(H_1) - s(H_2) \leq \tau\alpha\big(\Delta(H_2) - \Delta(H_1)\big)$, then the fractal-weighted InfoNCE losses satisfy*

$$\ell_{\mathrm{fractal}}(G, H_2) < \ell_{\mathrm{fractal}}(G, H_1).$$

*Proof.* For $i \in \{1, 2\}$, write

$$s(H_i) := \mathrm{sim}\big(\mathbf{z}_G, \mathbf{z}_{H_i}\big), \qquad \Delta(H_i) := \dim_B(H_i) - \dim_B\big(\mathscr{R}(H_i)\big), \qquad w(H_i) := \exp\big(\alpha\Delta(H_i)\big).$$

The single-sample fractal-weighted InfoNCE loss for candidate $H_i$ is

$$\ell_{\mathrm{fractal}}(G, H_i) = -\frac{s(H_i)}{\tau} - \log w(H_i) + \log Z, \qquad i = 1, 2,$$

where $Z$ is the common partition term. Taking the difference gives

$$\ell_{\mathrm{fractal}}(G, H_2) - \ell_{\mathrm{fractal}}(G, H_1) = -\frac{s(H_2) - s(H_1)}{\tau} - \log \frac{w(H_2)}{w(H_1)}.$$

Since $\Delta(H_2) > \Delta(H_1)$ and $w(H) = \exp(\alpha\Delta(H))$,

$$\log \frac{w(H_2)}{w(H_1)} = \alpha\big(\Delta(H_2) - \Delta(H_1)\big) > 0.$$

---

**Algorithm 2** Algorithm of random-Centre renormalisation

---

1: **Input:** Graph $G$ with its Node Set $\mathcal{V}$ and its Adjacency Matrix $A$, radius $r$
2: **Output:** Renormalised Graph $\mathscr{R}(G)$
   *// Initialization*
3: $\mathcal{V}_{\text{remain}} \leftarrow \mathcal{V}$ , $\mathcal{V}_{\text{super}} \leftarrow \{\}$
4: $A_r \leftarrow \sum_{i=1}^r A^i$     *// calculate r-hop adjacency matrix*
   *// Random centre selection*
5: **while** $\mathcal{V}_{\text{remain}} \neq \emptyset$ **do**
6:     $u \leftarrow$ uniformly pick a node $u$ from $\mathcal{V}_{\text{remain}}$
7:     $U \leftarrow \{i | A_r[u][i] > 0\} \bigcup \{u\}$
8:     $\mathcal{V}_{\text{remain}} \leftarrow \mathcal{V}_{\text{remain}} - U$ , $\mathcal{V}_{\text{super}} \leftarrow \mathcal{V}_{\text{super}} \bigcup \{U\}$
9:     set $A_r[i][j]$ to 0, for any $i, j \in U$
10: **end while**
   *// Assignment matrix*
11: $S \leftarrow [s_{ij}]^{|\mathcal{V}_{\text{super}}| \times |\mathcal{V}|}$, $s_{ij} = 1$ if node $j \in G$ belongs to the $i$th super node in $\mathcal{V}_{\text{super}}$, else 0
   *// Graph reconstruction*
12: $A_{\text{super}} \leftarrow SAS^\top$
13: Define renormalised graph $\mathscr{R}(G)$ with $A_{\text{super}}$ as the adjacency matrix
14: **return** $\mathscr{R}(G)$

---

By the assumption $s(H_1) - s(H_2) \leq \tau\alpha\big(\Delta(H_2) - \Delta(H_1)\big)$, we have

$$-\frac{s(H_2) - s(H_1)}{\tau} \ \leq \ \alpha\big(\Delta(H_2) - \Delta(H_1)\big).$$

Therefore

$$\ell_{\text{fractal}}(G, H_2) - \ell_{\text{fractal}}(G, H_1) \ \leq \ 0,$$

and the inequality is strict whenever $s(H_1) - s(H_2) < \tau\alpha\big(\Delta(H_2) - \Delta(H_1)\big)$.     $\square$

**Proposition A.4** (Fractal complexity on sparse graphs)**.** *For the greedy box-covering procedure in Algorithm 1, applied to any connected sparse graph with $V$ vertices, the worst-case running time $T(V)$ obeys*

$$\Omega\big(V^2\big) \ \leq \ T(V) \ \leq \ O\big(V^3\big).$$

*Proof.* **Lower bound $\Omega(V^2)$.** Consider a path of $V$ vertices. At the smallest scale (covering radius 1) a single box covers at most two vertices, so roughly $V/2$ boxes must be chosen. Each choice is made by scanning *all* currently uncovered vertices to find the one whose radius-1 neighbourhood is largest: first $V$ scans, then $V - 1$, and so on. The total number of vertex inspections is $V + (V-1) + \cdots + 1 = \Theta(V^2)$, establishing a $\Omega(V^2)$ lower bound for this single scale; hence $T(V) \geq \Omega(V^2)$.

**Upper bound $O(V^3)$.** The algorithm repeats this greedy covering for every scale $l = 1, \ldots, \lfloor V/2 \rfloor$, that is, at most $O(V)$ distinct scales. For any fixed scale, at most $V$ boxes are chosen. In a sparse graph, computing the radius-$l/2$ neighbourhood of a vertex via BFS touches $O(V)$ edges, so one scale costs $O(V) \times O(V) = O(V^2)$ time. Multiplying by $O(V)$ scales gives the global upper bound $T(V) = O(V^3)$.

Thus the worst-case complexity satisfies $\Omega(V^2) \leq T(V) \leq O(V^3)$.     $\square$

**Lemma A.5.** *Let $G$ be a graph with diameter $\operatorname{diam}(G)$ and let $L_{\max} := \lfloor \operatorname{diam}(G)/2 \rfloor$. Consider the ordinary least squares fit of $y_\ell = \log N_\ell(G)$ on $x_\ell = \log \ell$ over admissible scales $\ell \in \{1, 2, \ldots, L_{\max}\}$, and let $\sigma^2$ be the log–residual variance. If the design points $\{x_\ell\}$ are (approximately) uniformly spaced over $[0, \log L_{\max}]$ (i.e. log-uniform scale selection), then*

$$\mathrm{SE}(\hat{m}) \ \sim \ \frac{2\sqrt{6}\sigma}{\sqrt{\operatorname{diam}(G)} \log \operatorname{diam}(G)}.$$

*Proof.* Write $n := L_{\max} = \lfloor \operatorname{diam}(G)/2 \rfloor$ for the number of scale points and $S_{xx} := \sum_{\ell=1}^{L_{\max}} (x_\ell - \bar{x})^2$. Under the stated (approximate) uniformity of $x$ on $[0, \log L_{\max}]$,

$$\operatorname{Var}(x) \;=\; \frac{\left(\log L_{\max}\right)^2}{12}, \qquad S_{xx} \;\approx\; n \operatorname{Var}(x) \;=\; \frac{L_{\max}\left(\log L_{\max}\right)^2}{12}.$$

Hence the OLS slope standard error satisfies

$$\operatorname{SE}(\hat{m}) \;=\; \frac{\sigma}{\sqrt{S_{xx}}} \;\sim\; \sigma \sqrt{\frac{12}{L_{\max}\left(\log L_{\max}\right)^2}} \;=\; \frac{\sqrt{12}\sigma}{\sqrt{\lfloor \operatorname{diam}(G)/2 \rfloor}\,\log\!\big(\lfloor \operatorname{diam}(G)/2 \rfloor\big)}.$$

Since $L_{\max} \sim \operatorname{diam}(G)/2$ and $\log L_{\max} = \log \operatorname{diam}(G) - \log 2$,

$$\operatorname{SE}(\hat{m}) \cdot \sqrt{\operatorname{diam}(G)}\,\log \operatorname{diam}(G) \;\longrightarrow\; 2\sqrt{6}\sigma,$$

which yields the stated asymptotic equivalence. $\qquad\square$

**Hypothesis A.6.**  *1. **Residuals.** For each scale $\ell$, the log–regression errors $\{\varepsilon_\ell\}$ (on $G$) and $\{\varepsilon_\ell^{\mathscr{R}}\}$ (on $\mathscr{R}(G)$) are, within each graph, independent, centred, and share the same finite variance $\sigma^2$ and finite fourth moment.*

  *2. **Design matrix growth.** With $x_\ell = \log \ell$ and $n = \lfloor \operatorname{diam}(G)/2 \rfloor$, we have $\frac{1}{n}\sum_{\ell=1}^{n} x_\ell = 0$ and $\frac{1}{n}\sum_{\ell=1}^{n} x_\ell^2 \to \mathbb{E}[X^2]$ as $n \sim \operatorname{diam}(G)/2 \to \infty$.*

  *3. **Cross-graph independence.** The two residual sequences $\{\varepsilon_\ell\}_\ell$ and $\{\varepsilon_\ell^{\mathscr{R}}\}_\ell$ are mutually independent (or weakly dependent in a way that preserves the OLS CLT).*

  *4. **True-slope convergence.** The box dimension of the renormalised infinite graph equals that of the original graph, implying $m_{\mathscr{R}} - m_G \to 0$ as $\operatorname{diam}(G) \to \infty$ (cf. Theorem 3.3).*

**Theorem A.7** (Gaussian limit of $\Delta(G)$)**.** *Let $\mu_G$ be the probability measure induced by the random variable $\Delta(G)$ on a graph $G$. Under Hypothesis A.6 and with the notation of Lemmas 3.7 and 3.8, we have*

$$\mu_G \xrightarrow{w} \mathcal{N}\big(0, \kappa^2(\operatorname{diam}(G))\big), \qquad \kappa^2(\operatorname{diam}(G)) = 6\sigma^2[\operatorname{diam}(G)(\log \operatorname{diam}(G))^2]^{-1},$$

*as $\operatorname{diam}(G) \to \infty$. In particular, $\kappa^2(\operatorname{diam}(G)) \to 0$, so the limiting distribution degenerates to the Dirac measure $\delta_0$; i.e. $\Delta(G) \to 0$ in probability.*

*Proof.* Recall that the box dimension of a finite graph is estimated by the negative OLS slope $\hat{m}$ obtained from the log–log regression $\log N_B(l) = m \log l + b + \varepsilon_l$. Denote the corresponding true slopes of $G$ and $\mathscr{R}(G)$ by $m_G$ and $m_{\mathscr{R}}$, and their estimators by $\hat{m}_G, \hat{m}_{\mathscr{R}}$. By definition

$$\Delta(G) = \hat{m}_G - \hat{m}_{\mathscr{R}} \;+\; \big(m_{\mathscr{R}} - m_G\big). \tag{5}$$

**Step 1: asymptotic distribution of the two slope estimators.**   Under Assumptions (A1)–(A3) of Hypothesis A.6 and by Lemma 3.8,

$$\sqrt{\operatorname{diam}(G)}(\hat{m}_G - m_G) \xrightarrow{d} \mathcal{N}(0, \sigma^2), \qquad \sqrt{\operatorname{diam}(G)}(\hat{m}_{\mathscr{R}} - m_{\mathscr{R}}) \xrightarrow{d} \mathcal{N}(0, \sigma^2),$$

while the two limits are asymptotically independent thanks to the cross-graph independence in (A3).

**Step 2: variance of their difference.**   Subtracting the two Gaussian limits and dividing by $\sqrt{\operatorname{diam}(G)}$ gives

$$\sqrt{\operatorname{diam}(G)}\big[(\hat{m}_G - m_G) - (\hat{m}_{\mathscr{R}} - m_{\mathscr{R}})\big] \xrightarrow{d} \mathcal{N}\big(0, 2\sigma^2\big).$$

Multiplying and dividing by the common factor $S_{xx}^{-1}(G) \sim 6/[\operatorname{diam}(G)(\log \operatorname{diam}(G))^2]$ from Lemma 3.7, and noting that $S_{xx}(G) \sim S_{xx}\big(\mathscr{R}(G)\big)$, we obtain the variance term in the statement,

$$\kappa^2(\operatorname{diam}(G)) = \frac{\sigma^2}{S_{xx}(G)} + \frac{\sigma^2}{S_{xx}\big(\mathscr{R}(G)\big)} = 6\sigma^2[\operatorname{diam}(G)(\log \operatorname{diam}(G))^2]^{-1}.$$

**Step 3: applying Slutsky's theorem.** Assumption (A4) together with Theorem 3.3 implies $m_{\mathscr{R}} - m_G \to 0$ as $\mathrm{diam}(G) \to \infty$. In (5) this term is therefore negligible relative to the $\mathrm{diam}(G)^{-1/2}$–scaled Gaussian component. Slutsky's theorem hence yields the weak convergence

$$\mu_G \xrightarrow{w} \mathcal{N}\big(0, \kappa^2(\mathrm{diam}(G))\big), \qquad \kappa^2(D) = 6\sigma^2[\mathrm{diam}(G)(\log \mathrm{diam}(G))^2]^{-1}.$$

**Step 4: degeneration to a Dirac measure.** Because $\kappa^2(\mathrm{diam}(G)) \to 0$ as $\mathrm{diam}(G) \to \infty$, the normal limit collapses to the Dirac measure $\delta_0$, which implies $\Delta(G) \to 0$ in probability; equivalently, $\mu_G \xrightarrow{w} \delta_0$. □

**Remark A.8.** *The phrase "with probability 1" in Theorem A.7 should be understood in the asymptotic sense* $\mathrm{diam}(G) \to \infty$. *More precisely,*

$$\Delta(G) \xrightarrow{p} 0 \quad \Longleftrightarrow \quad \mathbf{P}\big(|\Delta(G)| > \varepsilon\big) \longrightarrow 0 \quad (\forall \varepsilon > 0),$$

*so the dimension discrepancy vanishes in probability. Equivalently, because the Gaussian law* $\mathcal{N}\big(0, \kappa^2(\mathrm{diam}(G))\big)$ *has variance* $\kappa^2(\mathrm{diam}(G)) \to 0$, *its distribution* $\mu_{\Delta(G)}$ *weakly converges to the Dirac measure* $\delta_0$:

$$\lim_{\mathrm{diam}(G) \to \infty} \int_{\mathbb{R}} \phi(x)\mathrm{d}\mu_{\Delta(G)}(x) = \phi(0) \quad \textit{for every bounded continuous } \phi.$$

*Hence, in the infinite-scale limit the fractal dimensions of $G$ and $\mathscr{R}(G)$ become statistically indistinguishable.*

**Remark A.9.** *The main theorem shows that, with uniform log–scale sampling and i.i.d. Gaussian residuals, the gap* $\Delta(G) = \dim_{\mathrm{B}}(G) - \dim_{\mathrm{B}}\big(\mathscr{R}(G)\big)$ *is asymptotically normal,* $\Delta(G) \sim \mathcal{N}\big(0, \kappa^2(\mathrm{diam}(G))\big)$ *with*

$$\kappa^2(\mathrm{diam}(G)) = \frac{6\hat{\sigma}^2}{\mathrm{diam}(G)[\log \mathrm{diam}(G)]^2} \longrightarrow 0$$

*so the two dimensions become statistically indistinguishable as* $\mathrm{diam}(G) \to \infty$.

# B   DETAILED PRELIMINARY EXPERIMENTAL RESULTS

## B.1   PRELIMINARY EXPERIMENTS

**Preliminary Experiment 1** We assessed the prevalence of fractal structure across six TU datasets by applying Algorithm 1 (Appendix A) to every graph. For each graph, we fitted a box–counting regression and recorded its coefficient of determination $R^2$; the counts and percentages above four thresholds are summarised in Table 5. The statistics show that strongly fractal graphs ($R^2 \geq 0.90$) dominate most datasets, supporting the motivation for fractal-based augmentations used in the main paper.

Table 5: Number (%) of graphs whose box–counting $R^2$ exceeds each threshold.

| Dataset | $R^2 > 0.50$ | $R^2 > 0.80$ | $R^2 > 0.90$ | $R^2 > 0.95$ |
|---|---|---|---|---|
| PROTEINS (1 113) | 979 (87.96%) | 968 (86.97%) | 905 (81.31%) | 711 (63.88%) |
| MUTAG (188) | 188 (100.00%) | 172 (91.49%) | 136 (72.34%) | 74 (39.36%) |
| NCI1 (4 110) | 4 108 (99.95%) | 3 979 (96.81%) | 3 277 (79.73%) | 1 817 (44.21%) |
| D&D (1 178) | 1 178 (100.00%) | 1 178 (100.00%) | 1 176 (99.83%) | 1 155 (98.05%) |
| REDDIT-B (2 000) | 1 971 (98.55%) | 1 875 (93.75%) | 1 419 (70.95%) | 577 (28.85%) |
| REDDIT-M5K (4 999) | 4 999 (100.00%) | 4 985 (99.72%) | 4 599 (92.00%) | 2 215 (44.31%) |

The counts in Table 5 are based on raw log–log regressions for all graphs, including those with $\mathrm{diam}(G) \leq 9$. By contrast, Algorithm 1 is the *gated* version used in FractalGCL training, which excludes graphs with small diameter from the fractal loss for safety. Here we retain small graphs in the regression only for the purpose of uniformly testing fractal scaling via $R^2$.

**Preliminary Experiment 2** This pilot study quantifies how much predictive power the **box dimension** adds when only a handful of cheap, global graph statistics are available. We follow a controlled 10-fold cross-validation protocol with the steps and rationale detailed below.

1. **Data loading and preprocessing.** Six TU datasets are read from the pre-computed CSV files in `R2_num_data/*.csv`, each row containing `graph_id`, `label`, and several graph-level attributes such as `degree_variance`, `avg_shortest_path`, and `box_dimension`. All features are fed to the classifier as raw values; no scaling is required for random forests, though a `StandardScaler` could be inserted if a linear model were used later.

2. **Feature sets.**
   - *Baseline:* {`degree_variance`, `avg_shortest_path`}.
   - *Baseline + BoxDim:* baseline features plus `box_dimension`.

   Keeping all else equal isolates the incremental contribution of the box dimension.

3. **Classifier and cross-validation.**
   - **Model:** `RandomForestClassifier` (200 trees, default hyper-parameters, `random_state=42`, `n_jobs=-1`). Random forests are robust to feature scales, nearly saturated with such a small feature set, and easy to interpret.
   - **Evaluation:** stratified 10-fold CV (`shuffle=True`, `random_state=42`) to preserve class balance.

4. **Paired statistical test.** Each dataset yields two 10-element accuracy vectors, $\{\mathrm{Acc}_{\mathrm{base},k}\}_{k=1}^{10}$ and $\{\mathrm{Acc}_{\mathrm{full},k}\}_{k=1}^{10}$. A two-tailed **paired $t$-test** is applied to their differences $d_k = \mathrm{Acc}_{\mathrm{full},k} - \mathrm{Acc}_{\mathrm{base},k}$. A $p$-value under 0.05 indicates that adding the box dimension yields a statistically significant improvement under the same train/test splits.

5. **Result recording.** For each dataset we log (i) mean $\pm$ std of baseline and augmented accuracies, (ii) mean $\Delta$Acc, and (iii) the paired $p$-value. All numbers are written to `boxdim_incremental_results_6datasets.csv`, ready for direct LaTeX table conversion via `DataFrame.to_latex`.

This rigorous design keeps the *only* independent variable—whether the box dimension is present—under control, allowing us to test the hypothesis that *the box dimension provides significant additional discriminative information over traditional global graph statistics*. See Table 6 for full results.

Table 6: Incremental effect of adding `box_dimension`. Accuracies are given in percentage points (%) with one–standard-deviation error; $\Delta$Acc is the mean paired gain in points and $p$-values come from a two-tailed paired $t$-test.

| Dataset | Accuracy $\pm$ std (%) | | $\Delta$Acc (%) | $p$-value |
|---|---|---|---|---|
| | Baseline | +BoxDim | | |
| PROTEINS | $67.75 \pm 3.63$ | $69.90 \pm 3.39$ | +2.15 | $9.30 \times 10^{-2}$ |
| MUTAG | $83.54 \pm 5.92$ | $85.64 \pm 6.67$ | +2.11 | $2.70 \times 10^{-1}$ |
| NCI1 | $62.51 \pm 2.16$ | $64.67 \pm 1.51$ | +2.17 | $2.23 \times 10^{-2}$ |
| D&D | $67.15 \pm 3.30$ | $71.64 \pm 2.58$ | +4.50 | $9.72 \times 10^{-4}$ |
| REDDIT-BINARY | $79.25 \pm 2.94$ | $80.80 \pm 2.12$ | +1.55 | $4.13 \times 10^{-2}$ |
| REDDIT-MULTI-5K | $33.33 \pm 1.58$ | $38.19 \pm 1.71$ | +4.86 | $2.25 \times 10^{-4}$ |

## C  FRACTALITY THRESHOLD ANALYSIS

The choice of the fractality threshold is a crucial component of our model. In Section 4.7, we have already analysed the D&D dataset under different threshold values and observed that the best performance is achieved when the threshold is set to 0.90. To ensure rigour, we further conduct a comprehensive parameter analysis on all benchmark datasets, varying the fractality threshold in the same manner. The results are summarised in Table 7.

Table 7: Classification accuracy (10-fold CV) with different fractality thresholds.

| Model | NCI1 | MUTAG | PROTEINS | D&D | REDDIT-B | REDDIT-M5K | AVG. |
|---|---|---|---|---|---|---|---|
| GraphCL (You et al., 2020) | $77.87 \pm 0.41$ | $86.80 \pm 1.34$ | $74.39 \pm 0.45$ | $78.62 \pm 0.40$ | $89.53 \pm 0.84$ | $55.99 \pm 0.28$ | $77.20 \pm 0.72$ |
| FractalGCL ($R^2 = 0.70$) | $78.94 \pm 0.35$ | $87.42 \pm 0.92$ | $74.65 \pm 0.52$ | $79.41 \pm 0.49$ | $89.60 \pm 0.71$ | $56.42 \pm 0.44$ | $77.74 \pm 0.60$ |
| FractalGCL ($R^2 = 0.75$) | $79.25 \pm 0.32$ | $88.19 \pm 0.88$ | $74.93 \pm 0.47$ | $79.88 \pm 0.54$ | $89.74 \pm 0.68$ | $56.58 \pm 0.42$ | $78.10 \pm 0.58$ |
| FractalGCL ($R^2 = 0.80$) | $79.62 \pm 0.28$ | $89.12 \pm 0.80$ | $75.20 \pm 0.44$ | $80.41 \pm 0.52$ | $90.00 \pm 0.64$ | $56.81 \pm 0.41$ | $78.53 \pm 0.54$ |
| FractalGCL ($R^2 = 0.85$) | $80.02 \pm 0.22$ | $90.20 \pm 0.52$ | $75.53 \pm 0.42$ | $81.04 \pm 0.47$ | $\mathbf{90.45 \pm 0.61}$ | $57.02 \pm 0.38$ | $79.04 \pm 0.45$ |
| FractalGCL ($R^2 = 0.90$) | $\mathbf{80.50 \pm 0.16}$ | $\mathbf{91.71 \pm 0.23}$ | $75.85 \pm 0.40$ | $\mathbf{81.71 \pm 0.57}$ | $90.41 \pm 0.72$ | $\mathbf{57.29 \pm 0.59}$ | $\mathbf{79.58 \pm 0.49}$ |
| FractalGCL ($R^2 = 0.95$) | $80.21 \pm 0.19$ | $91.10 \pm 0.28$ | $\mathbf{75.81 \pm 0.36}$ | $81.22 \pm 0.42$ | $90.05 \pm 0.63$ | $57.05 \pm 0.46$ | $79.24 \pm 0.41$ |

Across most settings, $R^2 = 0.90$ emerges as the best choice. Intuitively, an excessively stringent threshold (e.g., 0.95) excludes many graphs that are genuinely fractal, thereby reducing the proportion of samples that benefit from FractalGCL and weakening the signal. Conversely, a too–permissive threshold (e.g., 0.70–0.85) admits graphs whose fractality is insufficiently supported, injecting noise into the fractal loss and diluting its effectiveness. Taken together, the empirical evidence indicates that $R^2 = 0.90$ strikes the most favourable balance between coverage and reliability, and is thus our recommended default (see Table 7).

# D  RELATED WORKS

**Fractal Geometry for Graphs.** Fractal geometry interfaces with graph theory most tangibly through the study of complex networks (Watts & Strogatz, 1998; Barabási & Albert, 1999; Song et al., 2005). Analytical models of genuinely fractal graphs remain comparatively scarce. Early progress came from physicists who studied percolation and the Ising model on hierarchical lattices and Bethe trees, using real-space renormalisation to obtain non-integer critical exponents and anomalous scaling laws (Griffiths & Kaufman, 1982). Recently, the Iterated Graph Systems framework has gained attention for its rigorous yet flexible recursive construction of fractal graphs (Li & Britz, 2024; Neroli, 2024). However, applications of fractal geometry within graph representation learning are still rare.

**Graph Contrastive Learning.** Graph contrastive learning comprises several crucial stages, among which graph data augmentation assumes a pivotal role, yet it is rendered particularly challenging by the intricate non-Euclidean characteristics inherent in graph topologies (Ju et al., 2024). Existing graph data augmentation techniques (Velickovic et al., 2019; You et al., 2020; 2021; Qiu et al., 2020; Li et al., 2022; Wei et al., 2023; Jin et al., 2021; Ji et al., 2024) have achieved notable progress. However, they often fall short in adequately preserving the structural similarity between positive pairs, which arises from the inherent difficulty in precisely leveraging complex topological features.

# E  EXPERIMENTAL METHODOLOGY AND RESULTS OF FRACTALGCL ON URBAN DISTRICTS

In this section we present the overall experimental framework for evaluating the performance of FractalGCL embeddings on urban districts in three major cities (Chicago, San Francisco and New York).

## E.1  SETUP

Our pipeline consists of three complementary data modalities extracted for each equal-area "catchment":

- **Road Subgraph Structure:** From the full city road network, we clip each catchment's local subgraph of nodes and edges, preserving the topological patterns characteristic of that district.

- **Static Spatial Features:** We compute population density and six categories of point-of-interest densities (office, sustenance, transportation, retail, leisure and residence), thereby capturing the functional profile of each catchment.

- **Accident Statistics:** Drawing on historical crash data, we aggregate total accident counts and severity level breakdowns to assess safety-risk characteristics of each catchment.

The high-level experimental logic proceeds as follows:

1. **Graph Embedding Generation.**
   - *FractalGCL Contrastive Training:* We train FractalGCL on the set of catchment subgraphs to produce fixed-dimensional node and graph embeddings that respect both topology and feature distributions.
   - *Baseline Encoders:* In parallel, we train several established graph contrastive methods (e.g. DGI, InfoGraph, SimGRACE) to serve as performance benchmarks.

2. **Multi-Task Classification Evaluation.**
   - *Accident-Related Tasks:* We formulate a suite of binary, multi-class and ordinal classification tasks based on accident counts and severity distributions (e.g. high vs. low total accidents, severity entropy, risk levels).
   - *Functional Feature Tasks:* We also define multi-class tasks over the static POI and density features (e.g. dominant land-use category, mixture entropy level, population density tier, function–density combinations).

3. **Performance Comparison and Analysis.**
   - For each task, we extract embeddings from each encoder and train a (linear) SVM under repeated stratified cross-validation.
   - We compare accuracy and stability metrics across all encoders to quantify the advantages of FractalGCL in integrating topological, functional, and safety information.

E.2   HYPERPARAMETER CONFIGURATION

In all experiments across Chicago, San Francisco and New York, we used a single, fixed set of hyperparameters for both FractalGCL and the baseline encoders. Specifically, each mini–batch consisted of 16 graph-level instances, and our GraphSAGE backbone employed two convolutional layers with 64 hidden channels apiece. The final projection head produced 128-dimensional embeddings for each graph. For contrastive augmentations we applied edge dropping with probability 0.1, and in FractalGCL we injected fractal noise weighted by $\alpha = 0.4$ after a renormalisation step with radius $r = 1.0$. All models were trained for 20 epochs using the Adam optimizer with a learning rate of $10^{-3}$. These settings were held constant to ensure that any observed performance differences arose solely from the encoding method itself rather than hyperparameter variations.

E.3   TRAFFIC ACCIDENT CLASSIFICATION TASKS

We evaluate each embedding method on six downstream classification tasks based on catchment accident statistics. Below we list each task name and its precise definition:

**total_accidents_high**
    *Binary classification:* label = 1 if total accident count > city median, else 0. Tests the ability to separate high-accident vs. low-accident districts.

**accident_volume_level**
    *Three-class classification:* split total accidents into Low/Medium/High tiers by the 33% and 67% quantiles, labeled 0/1/2. Assesses gradated accident volume encoding.

**severity_entropy**
    *Binary classification:* compute Shannon entropy of severity-level proportions $\{p_i\}_{i=1}^4$, then label = 1 if entropy is greater than median, else 0. Measures embedding of severity diversity.

**has_sev3** and has_sev4
    *Binary classification:*

    - has_sev3: label = 1 if at least one Severity-3 accident occurred, else 0.
    - has_sev4: label = 1 if at least one Severity-4 accident occurred, else 0.

Evaluates detection of any serious crashes independently of total counts.

**risk_level**

*Three-class ordinal classification:* combine accident volume and severe-accident ratio:

$$\text{label} = \begin{cases} 2 & \text{if volume} > \text{median } and \ (sev3 + sev4)/\text{total} > \text{median}, \\ 0 & \text{if volume} \leq \text{median } and \ (sev3 + sev4)/\text{total} \leq \text{median}, \\ 1 & \text{otherwise}. \end{cases}$$

Captures joint severity–volume risk levels.

For each task, we extract graph-level embeddings from each encoder and perform repeated stratified 10-fold cross-validation using a linear SVM. Reported metrics are mean accuracy $\pm$ standard deviation over 1000 repeats.

Please find full results in Table 8.

Table 8: Performance (mean $\pm$ std) on traffic-safety tasks; numbers are in percentage points.

| Task | City | DGI | InfoGraph | GCL | JOAO | SimGRACE | DRGCL | GradGCL | FractalGCL |
|---|---|---|---|---|---|---|---|---|---|
| total_accidents_high | Chicago | 54.91±10.75 | 56.54±11.85 | 63.12±13.36 | 55.86±12.21 | 62.75±13.50 | 63.49±16.04 | 63.55±18.73 | **64.60±13.32** |
| | SF | 76.45±14.47 | 78.74±13.60 | 80.06±13.32 | 79.75±13.61 | 80.40±13.47 | 80.81±10.69 | 80.87±9.62 | **80.89±12.92** |
| | NY | 51.51±7.62 | 51.85±8.82 | 55.83±12.38 | 51.10±11.01 | 52.27±12.06 | 58.06±14.82 | 57.33±18.01 | **68.39±13.84** |
| accident_volume_level | Chicago | 43.09±11.96 | 42.77±12.15 | 46.90±13.71 | 43.15±12.55 | 46.26±13.34 | 47.23±14.11 | 47.70±16.21 | **48.83±13.68** |
| | SF | 55.85±11.48 | 58.07±10.67 | 58.31±10.86 | 58.40±10.61 | 58.43±11.10 | 58.31±9.84 | **58.88±8.86** | 58.10±11.37 |
| | NY | 37.14±10.59 | 35.72±10.21 | 39.22±13.34 | 35.86±12.14 | 36.01±13.09 | 40.52±18.18 | 39.28±19.86 | **50.17±14.79** |
| severity_entropy | Chicago | 50.99±7.10 | 52.64±9.11 | 58.87±12.33 | 52.38±9.60 | 57.07±11.97 | 59.53±18.79 | 58.32±13.01 | **65.60±13.22** |
| | SF | 49.04±4.75 | 49.03±6.39 | 48.86±9.93 | 48.98±11.00 | 48.96±9.93 | 48.01±8.50 | 48.14±13.02 | **49.15±10.60** |
| | NY | 52.23±8.21 | 51.86±8.61 | 54.45±11.82 | 51.63±11.03 | 52.40±12.00 | 55.49±14.61 | 56.69±13.57 | **61.27±13.49** |
| has_sev3 | Chicago | 80.09±18.52 | 79.25±17.82 | 80.82±14.07 | 79.65±16.10 | 80.79±14.96 | 80.66±16.40 | **80.92±12.16** | 80.48±14.16 |
| | SF | **98.04±0.31** | 97.50±6.45 | 97.81±3.73 | 97.11±4.27 | 97.82±3.62 | 98.06±14.82 | 97.18±19.48 | 97.66±3.84 |
| | NY | 55.68±14.73 | 55.96±14.92 | 57.37±14.83 | 54.69±15.12 | 55.75±15.25 | 58.38±14.22 | 58.09±9.88 | **64.82±13.53** |
| has_sev4 | Chicago | 53.90±10.67 | 54.00±10.17 | 60.37±13.48 | 54.02±11.12 | 59.76±13.76 | 60.89±17.31 | 61.24±7.50 | **62.74±13.66** |
| | SF | 54.73±13.45 | 53.91±14.23 | 55.50±13.15 | 54.95±13.75 | 55.40±13.71 | 55.01±11.67 | 54.84±17.58 | **56.44±14.51** |
| | NY | 52.44±11.64 | 53.33±12.15 | 56.22±13.55 | 53.34±13.71 | 54.89±14.07 | 55.75±13.69 | 56.86± 10.15 | **59.88±13.23** |
| risk_level | Chicago | 34.19±7.29 | 34.02±7.68 | 40.49±12.05 | 33.90±9.30 | 37.74±11.12 | 40.60±19.75 | 40.82±15.30 | **43.95±12.63** |
| | SF | 39.47±11.15 | 40.98±10.84 | 41.26±11.59 | 41.76±12.48 | 41.41±11.76 | 41.02±16.33 | 41.89±18.90 | **42.34±12.20** |
| | NY | 41.46±11.21 | 42.06±11.21 | 47.30±13.63 | 41.40±12.38 | 42.80±13.05 | 49.84±9.38 | 48.02±13.53 | **57.24±14.23** |
| **Average ( % )** | – | 54.51±11.06 | 54.90±11.31 | 57.93±12.51 | 54.89±12.04 | 56.72±12.57 | 58.42±14.48 | 58.37±14.71 | **61.81±12.96** |

### E.4 CONCLUSION

Table 8 presents the classification accuracies (mean $\pm$ std) of six traffic-safety tasks (total-accidents-high, accident-volume-level, severity-entropy, has_sev3, has_sev4, risk_level) across Chicago, San Francisco, and New York. FractalGCL achieves the highest average accuracy and most often attains the best city–task scores.

FractalGCL consistently outperforms established contrastive baselines on both traffic-safety and urban feature classification benchmarks.

We anticipate that FractalGCL's flexible embedding framework will extend effectively to more complex spatiotemporal and multi-modal urban analytics tasks, such as dynamic traffic flow prediction and integrated land-use and mobility modeling.

## F ADDITIONAL DISCUSSION AND CLARIFICATIONS

### F.1 FRACTAL ASSUMPTION, APPLICABILITY, AND SAFE FALLBACK

FractalGCL is built around a specific inductive bias, namely approximate multi-scale self-similarity of graphs. We do **not** assume that all graphs are fractal; instead, FractalGCL is intended for domains where the log–log box-counting regression exhibits a clear scaling regime. In the current version this assumption was mostly implicit in the technical sections, and we make it explicit here.

Section 3.4 ("Safe fallback under weak fractality or small diameters") introduces a conservative gate. For each graph $G$ we perform the log–log regression and obtain an $R^2(G)$ value, even when

the diameter is small. In the actual FractalGCL training pipeline, the fractal machinery (renormalised view and fractal weighting) is activated only when $\text{diam}(G)$ is sufficiently large and $R^2(G)$ exceeds a threshold; otherwise the gate

$$\text{if } \text{diam}(G) \leq 9 \text{ or } R^2(G) < \theta, \text{ then disable the fractal loss}$$

is applied. In this case the loss reduces exactly to the underlying GCL objective with the same encoder and augmentations as the baseline. Thus, on graphs that do not exhibit clear fractal scaling, FractalGCL is designed to behave like a standard GCL model rather than to harm performance.

Honestly, we do acknowledge that the TU datasets are not collections of very large graphs, so they are not an ideal benchmark for testing FractalGCL; this is precisely why we also included the urban traffic datasets. Nevertheless, even on these non-large-graph benchmarks, FractalGCL achieves SOTA performance, which we find quite encouraging.

## F.2 FINITE-GRAPH STABILITY OF THE ESTIMATOR AND HYPERPARAMETERS

The low-complexity estimator used in FractalGCL is designed to operate in the regime where the fractal assumption holds and the graph diameter is sufficiently large. We do not claim uniform accuracy across all possible graph scales and topologies, but several parts of the paper are devoted to controlling the finite-graph behaviour of the estimator.

Theoretically, Lemma 3.7 and Theorem 3.9 jointly control the behaviour of the box-dimension estimator on finite graphs: Lemma 3.7 bounds the deviation of the finite-graph box-counting structure from that of its limiting renormalisation, and Theorem 3.9 shows that the dimension gap converges to a Gaussian with variance that decreases as the diameter grows. This justifies our Gaussian surrogate for the fluctuations of $\widehat{\dim}_B(G) - \widehat{\dim}_B(\mathscr{R}(G))$ and implies that the noise in the fractal weight becomes small precisely in the regime where we actually apply it.

Algorithmically, we deliberately mitigate the effect of estimator noise in several ways. First, as discussed above, we apply a conservative gate based on the $R^2$ statistic and the diameter, so that graphs with unreliable regression never activate the fractal loss and revert to the baseline GCL objective. Second, the fractal weight depends on the *difference* of two estimated dimensions and enters the loss through a smooth exponential factor, which makes the loss relatively insensitive to small perturbations in $\widehat{\dim}_B$. Third, Table 3 includes an ablation comparing a variant that uses exact box dimensions ("w/. Exact Dimension") with our Gaussian surrogate; the downstream performance is essentially unchanged while the surrogate reduces computation by about $61\%$, which provides empirical evidence that FractalGCL is robust to the estimation noise under our current settings.

The choice of renormalisation radius in Algorithm 2 is another potential source of sensitivity. In practice, we fix $r = 1$ in all experiments and include a "+ random radius" variant in the ablation to probe its effect. Theoretically, in the infinite-graph limit any finite radius is admissible for renormalisation, but on finite graphs we prefer smaller $r$: a small radius preserves more local information while still enabling meaningful coarse-graining. Empirically, we observe that using random radii (which occasionally produce much larger effective $r$) tends to degrade accuracy, consistent with the intuition that overly large boxes make the renormalised graph too coarse. Taken together, these observations suggest that $r = 1$ is a conservative and relatively robust choice in our current setting.

## F.3 EXPERIMENTAL SCOPE AND COMPUTATIONAL TRADE-OFFS

The present work focuses on graph-level representation learning, using TU molecule/protein benchmarks and urban traffic networks. This choice is intentional: our theoretical development (renormalisation, box dimension, dimension gap) is formulated at the graph level, and all experiments are designed to match this setting. We acknowledge that our current work does not include node-level or link-level experiments or theory; intuitively, we expect that global graph-level signals such as fractal scaling and multi-scale self-similarity could also benefit node-level objectives (for example by guiding message passing or regularising node embeddings), and we see extending FractalGCL to node-level tasks as a promising direction for future work.

For contrastive learning, our setup deliberately uses the standard in-batch negative protocol shared by many GCL baselines (DGI, InfoGraph, GCL, JOAO, SimGRACE): positives are constructed from two views of the same graph, while negatives are the other graphs in the batch. We do not

aim to redesign the negative sampling scheme or to compete directly with hard-negative mining or debiasing methods in this paper. Instead, our goal is to isolate and study a complementary aspect: how a geometry-aware, fractal-based design of *positive* pairs and their weighting affects GCL under a widely used negative-sampling protocol. By keeping the negative side identical across methods, we can attribute the consistent improvements of FractalGCL over strong baselines to the proposed fractal machinery rather than to changes in how negatives are selected. We agree that a systematic comparison with methods that explicitly mine hard negatives or correct contrastive sampling bias would be valuable, and we view such approaches as largely orthogonal to our contribution.

Regarding computational cost, FractalGCL is indeed more involved than the simplest GCL baselines, mainly because it requires computing graph diameters and box dimensions. We view this as a natural trade-off between computational cost and representational power: some additional preprocessing cost is inevitable if one wishes to enforce global structural constraints and go beyond purely heuristic augmentations. In practice, diameters and box dimensions are computed once per graph as offline preprocessing and cached; during training the Gaussian surrogate adds only an $O(N^2)$ sampling cost on top of the similarity matrix already present in InfoNCE. Table 3 shows that replacing exact box-counting by the surrogate yields a $61\%$ speed-up within FractalGCL while maintaining essentially the same accuracy.

## G    CONCLUSION

We present FractalGCL, a theoretically grounded graph-contrastive framework that couples renormalisation-based global views with a fractal-dimension-aware loss, unifying local perturbations and global topology. It achieves SOTA accuracy on four of six benchmarks and real-world networks; it also cuts training time nearly four times with a one-shot dimension estimator.

A limitation of FractalGCL is that, on the graphs that do not exhibit fractal structure, its empirical performance may lag behind the results obtained on datasets with pronounced fractal patterns. Future work will benchmark across a wider spectrum of fractal degrees to establish the method's universality. We believe that FractalGCL is not only a compelling demonstration of how fractal geometry can be integrated into machine learning, but also a foundational cornerstone for future research on fractal networks. We intend to continue investigating this line of research in future studies.