# OpenReview forum: "Fractal Graph Contrastive Learning"
_ICLR.cc/2026/Conference — Submitted to ICLR 2026_

### Official Review · Reviewer_gMeF · 2025-10-22

**Soundness:** 3
**Presentation:** 2
**Contribution:** 1
**Rating:** 2
**Confidence:** 5

**Summary:**

This paper proposes FractalGCL, a theory-driven framework that enhances graph contrastive learning by explicitly maintaining global structural consistency through fractal-based augmentation. The method introduces a renormalisation-based augmentation and a fractal-dimension-aware contrastive loss, providing both improved representation quality and a theoretical performance lower bound even for non-fractal graphs. Moreover, the authors derive a one-shot estimator to efficiently approximate fractal dimensions, achieving training time reduction while attaining state-of-the-art performance across benchmark and traffic network datasets.

**Strengths:**

- The writing is good.
- The work is theoretically grounded.
- The performance is superior to the baselines.

**Weaknesses:**

* The overall design lacks consistency and coherence.
* The selection of baselines is not sufficiently comprehensive.
* There is performance inconsistency.
* The motivation experiment (Table 5) appears to contain methodological or statistical errors.

**Questions:**

* In the Introduction, the stated motivation focuses on ensuring *global semantic consistency* between positive pairs (inter-view similarity). However, the subsequent discussion emphasizes *self-similarity* and *hierarchical complexity*, both are intra-view properties. These two objectives operate at different conceptual levels and lack a unified logical connection.
* The renormalisation algorithm currently considers only graph topology. How can it be extended to handle graphs with node and/or edge attributes?
* In Section 3.2, the notation $z_n$ is overloaded: in *Mapping from Graph G to Representation z*, it refers to the embedding of the renormalised graph, while in *Contrastive Loss with Fractal Weight*, it denotes the embedding of the original graph. This inconsistency confuses the reader.
* In Section 3.2 and Figure 1, the augmented views are described as disjoint unions from two independent runs. However, in Section 3.3, the contrastive loss is defined between the original and renormalised graphs, not between two different unions. This discrepancy makes it difficult to assess the correctness and validity of the overall framework.
* The assumptions (A1–A4) referenced in Lemma 3.8 are not explicitly stated.
* What is the model’s performance under a transfer learning setting (e.g., MoleculeNet)?
* Please provide details on computational overhead (GPU memory usage and training time) compared to the baselines.
* What are the runtime complexities of Algorithms 1 and 2?
* In Table 4, the result for D&D (80.14) is notably lower than that in Table 1 (81.71). Please explain this inconsistency.
* A sensitivity analysis on the radius parameter used in the greedy box covering should be included.
* In Table 5, the authors report the number of graphs whose box-counting $R^2$ exceeds various thresholds. According to Algorithm 1, when the graph diameter ≤ 9, $R^2 = 0$. Based on my statistical check, the number of graphs with diameter > 9 defines the upper bound for Table 5 values. However, the reported ratios for $R^2 > 0.5$ exceed these bounds, raising serious concerns about the validity and authenticity of the results.


| Dataset    | #Graph | #Diameter<=9 | #Diameter>9 | Max-possible-$R^2>*$-ratio(%) |
|------------|--------|--------------|-------------|-----------------------------|
| PROTEINS   | 1113   | 521          | 592         | 53.19                       |
| MUTAG      | 188    | 143          | 45          | 23.94                       |
| NCI1       | 4110   | 889          | 3221        | 78.37                       |
| D&D        | 1178   | 31           | 1147        | 97.37                       |
| REDDIT-B   | 2000   | 910          | 1090        | 54.50                       |
| REDDIT-M5K | 4999   | 824          | 4175        | 83.52                       |


* Add more recent SOTAs, including TopoGCL[1], GCS[2], SEGA[3], CI-GCL[4], StructPosGSSL[5].



[1] Chen Y, Frias J, Gel Y R. TopoGCL: Topological graph contrastive learning[C]//Proceedings of the AAAI conference on artificial intelligence. 2024, 38(10): 11453-11461.

[2] Wei C, Wang Y, Bai B, et al. Boosting graph contrastive learning via graph contrastive saliency[C]//International conference on machine learning. PMLR, 2023: 36839-36855.

[3] Wu J, Chen X, Shi B, et al. Sega: Structural entropy guided anchor view for graph contrastive learning[C]//International Conference on Machine Learning. PMLR, 2023: 37293-37312.

[4] Tan S, Li D, Jiang R, et al. Community-Invariant Graph Contrastive Learning[C]//International Conference on Machine Learning. PMLR, 2024: 47579-47606.

[5] Wijesinghe A, Zhu H, Koniusz P. Graph Self-Supervised Learning with Learnable Structural and Positional Encodings[C]//Proceedings of the ACM on Web Conference 2025. 2025: 4053-4067.

---

> ### Author Response · Authors · 2025-11-20
> **Response to Reviewerg gMeF (i)**
>
> We sincerely thank reviewer gMeF for the thoughtful comments and questions, especially Question 1, 6 and 11, which are both deep and wide in scope.
> From the Questions, we clearly see the amount of careful reading and serious reflection that the reviewer has devoted to our work.
> These suggestions and questions are crucial for improving the paper. Below we provide our detailed responses to the Questions.
>
> 1. *Question: In the Introduction, the stated motivation focuses on ensuring global semantic consistency between positive pairs (inter-view similarity). However, the subsequent discussion emphasizes self-similarity and hierarchical complexity, both are intra-view properties. These two objectives operate at different conceptual levels and lack a unified logical connection.*
>
>    **Response to Question 1.**
>    We thank the reviewer for raising this point.
>    Our objective is indeed to enforce global semantic consistency between positive pairs in the contrastive framework.
>    The discussion of self-similarity and hierarchical complexity in the Introduction is intended to provide the mechanism by which this global consistency is achieved.
>
>    More specifically, the fractal structure is an intra-graph property: a fractal graph exhibits statistically similar structure across scales, and its global geometry can be captured by its box-counting dimension.
>    The renormalisation procedure constructs, from a given graph $G$, a renormalised graph $\mathscr{R}(G)$ that preserves this global fractal structure (Theorem 3.3).
>    Thus, $G$ and $\mathscr{R}(G)$ are two views that share the same multi-scale geometric pattern, even though they differ at the level of individual nodes and edges.
>    In our method, these two graphs form a positive pair in the contrastive objective and, consequently, the intra-view fractal property is used precisely to guarantee inter-view structural alignment: we exploit fractal self-similarity as an inductive bias to ensure that augmented views preserve global graph geometry.
>
>    We will revise the Introduction to make this logical connection explicit and to clearly explain how self-similarity and hierarchical complexity provide a unified mechanism for enforcing global semantic consistency between views.
>
> 2. *Question: The renormalisation algorithm currently considers only graph topology. How can it be extended to handle graphs with node and/or edge attributes?*
>
>    **Response to Question 2.**
>    The reviewer raises a very good point about attributed graphs.
>    The approach for attributed graphs was considered at the beginning, but in the present manuscript, we deliberately formulate the renormalisation procedure in purely topological terms, because an explicit treatment of general node and edge attributes would substantially complicate the notation and obscure the main ideas.
>    Nevertheless, the construction extends in an accessible way to attributed graphs.
>
>    For node attributes, when a box $U_i$ is formed in the greedy covering, one can assign to the corresponding supernode a feature vector obtained by aggregating the features of the original nodes in $U_i$. A standard choice is to use a permutation-invariant pooling operator (for example mean, sum, or max pooling); for categorical node labels one may first apply a one-hot encoding and then use the same aggregation.
>    For edge attributes, whenever a superedge between boxes $U_i$ and $U_j$ is created, its feature vector can be obtained by aggregating the attributes of all edges with one endpoint in $U_i$ and the other in $U_j$, again using a permutation-invariant pooling operator.
>
>    In the revised version we will add a brief remark to Section 3.2 to clarify this extension.
>
> 3. *Question: In Section 3.2, the notation $z_n$ is overloaded: in Mapping from Graph G to Representation z, it refers to the embedding of the renormalised graph, while in Contrastive Loss with Fractal Weight, it denotes the embedding of the original graph. This inconsistency confuses the reader.*
>
>    **Response to Question 3.**
>    We thank the reviewer for pointing out the notational inconsistency in Section 3.2.
>    Our intention throughout is to encode both the original graph $G_n$ and its renormalised counterpart $\mathscr{R}(G_n)$ with a shared encoder, obtain graph-level representations for each of them, and then apply the contrastive loss to this positive pair.
>    The overloading of the symbol $z_n$ in the current text does not reflect this intended design and can indeed be confusing.
>    In the implementation, the mapping from a graph to its representation is applied separately to $G_n$ and $\mathscr{R}(G_n)$, and the resulting embeddings are then used in the contrastive loss.
>
>    To remove this ambiguity in the manuscript, we will revise Section 3.2 so that the representation of the original graph is consistently denoted by $z_n$, while the representation of the renormalised graph is denoted by $z_n^{\mathrm{(R)}}$.

---

> > ### Author Response · Authors · 2025-11-20
> > **Response to Reviewerg gMeF (ii)**
> >
> > 4. *Question: In Section 3.2 and Figure 1, the augmented views are described as disjoint unions from two independent runs. However, in Section 3.3, the contrastive loss is defined between the original and renormalised graphs, not between two different unions. This discrepancy makes it difficult to assess the correctness and validity of the overall framework.*
> >
> >    **Response to Question 4.**
> >    We very much appreciate the reviewer pointing out this apparent discrepancy.
> >    The current draft does not clearly separate (i) how augmented views are constructed at the graph
> >    level and (ii) how the contrastive loss is defined on the resulting representations, which can indeed
> >    be confusing.
> >
> >    Conceptually, the positive pair is always derived from a single original graph $G_n$ and its
> >    renormalised counterpart $\mathscr{R}(G_n)$.
> >    In practice, we first generate two augmented views
> >    $\tilde G_n^{(1)}$ and $\tilde G_n^{(2)}$ from this pair and then apply the GNN encoder and the
> >    fractal-weighted InfoNCE loss.
> >    Section 3.2 focuses on how one such view can be constructed using the disjoint union
> >    $G_n \sqcup \mathscr{R}(G_n)$, while Section 3.3 presents the loss in terms of the underlying pair
> >    $(G_n, \mathscr{R}(G_n))$.
> >    Thus, the disjoint union is a concrete design choice for building an augmented view from
> >    $(G_n, \mathscr{R}(G_n))$, rather than a different notion of “positive pair”.
> >
> >    More concretely, in the implementation we encode two views
> >    $\tilde G_n^{(1)}$ and $\tilde G_n^{(2)}$ obtained from $G_n$ and $\mathscr{R}(G_n)$ via local
> >    augmentations, and we compute the loss on their embeddings
> >    $z_n^{(1)} = g_\phi(\mathrm{Readout}(f_\theta(\tilde G_n^{(1)})))$ and
> >    $z_n^{(2)} = g_\phi(\mathrm{Readout}(f_\theta(\tilde G_n^{(2)})))$.
> >    Whether $\tilde G_n^{(i)}$ is realised as $G_n \sqcup \mathscr{R}(G_n)$ or as two separate graphs fed
> >    through the shared encoder is a design choice, and Table 3 explicitly compares the variant
> >    “w/o Graph Concat”, where unions are not used at all.
> >    In all cases, the fractal weights are computed from the pair $(G_n, \mathscr{R}(G_n))$.
> >
> >    We will revise Sections 3.2–3.3 and Figure 1 to explicitly introduce the two augmented views
> >    $\tilde G_n^{(1)}$ and $\tilde G_n^{(2)}$, to state clearly when a disjoint union
> >    $G_n \sqcup \mathscr{R}(G_n)$ is used as an input to the encoder, and to make explicit that the
> >    contrastive loss is always defined on two views derived from the same pair $(G_n, \mathscr{R}(G_n))$.
> >    This will remove the ambiguity between the view construction and the loss definition while
> >    keeping the framework unchanged.
> >
> > 5. *Question: The assumptions (A1–A4) referenced in Lemma 3.8 are not explicitly stated.*
> >
> >    **Response to Question 5.**
> >    We thank the reviewer for drawing attention to this missing cross-reference.
> >    The four conditions used in Lemma 3.8 are exactly the four items collected under
> >    *Hypothesis A.6* in Appendix A, namely: independent centred residuals with finite variance and
> >    fourth moment; growth of the log-scale design matrix; cross-graph independence of residuals; and
> >    convergence of the true slopes implied by Theorem 3.3.
> >
> >    In the revised version we will simply replace “under Assumptions (A1)-(A4)” by “under Hypothesis A.6”.

---

> > > ### Author Response · Authors · 2025-11-20
> > > **Response to Reviewerg gMeF (iii)**
> > >
> > > 6. **Question:** What is the model’s performance under a transfer learning setting (for example, MoleculeNet)?
> > >
> > >    **Response to Question 6.**
> > >    We thank the reviewer for proposing evaluating FractalGCL in a transfer-learning setting.
> > >    Due to time limitation in the rebuttal period, we conducted a cross-city transfer study (rather than a MoleculeNet study) on the three urban traffic datasets (New York, Chicago and San Francisco).
> > >
> > >    Concretely, we first train FractalGCL on the New York data and then reuse the resulting encoder in the other cities; we denote this variant by **FractalGCL(NY)**.
> > >    The table below reports the classification accuracies (mean $\pm$ standard deviation) for six accident-related tasks across the three cities.
> > >    Averaged over all 18 tasks, FractalGCL(NY) achieves $60.91\pm13.26$, which clearly improves over the strongest non-fractal baseline (SimGRACE, $56.64\pm14.85$) and is close to the fully in-domain FractalGCL trained separately on each city ($62.04\pm12.49$).
> > >    On the source city (New York), FractalGCL(NY) attains the best performance, and on the target cities (Chicago and San Francisco) it typically matches or exceeds the best baselines while being only a few percentage points below the in-domain FractalGCL.
> > >    These results indicate that the fractal representations learned by FractalGCL may transfer across different cities in a realistic cross-domain setting.
> > >
> > >
> > >
> > >    | Task                     | City    | DGI                | InfoGraph          | GCL                | JOAO               | SimGRACE           | FractalGCL(NY)       | FractalGCL           |
> > >    |--------------------------|---------|--------------------|--------------------|--------------------|--------------------|--------------------|----------------------|----------------------|
> > >    | `total_accidents_high`   | Chicago | $57.92\pm11.57$    | $59.26\pm12.84$    | $63.24\pm14.41$    | $61.11\pm13.32$    | $61.82\pm14.48$    | $63.35\pm13.40$      | $65.48\pm14.66$      |
> > >    | `total_accidents_high`   | SF      | $77.28\pm13.23$    | $78.23\pm12.18$    | $78.69\pm11.93$    | $78.60\pm13.00$    | $79.01\pm12.03$    | $78.27\pm12.27$      | $79.02\pm12.00$      |
> > >    | `total_accidents_high`   | NY      | $52.88\pm7.61$     | $49.49\pm6.89$     | $51.20\pm11.61$    | $49.94\pm10.91$    | $49.84\pm11.89$    | $68.17\pm13.86$      | $66.40\pm14.07$      |
> > >    | `accident_volume_level`  | Chicago | $42.67\pm12.00$    | $42.73\pm12.36$    | $45.23\pm13.12$    | $42.38\pm12.17$    | $44.42\pm13.14$    | $46.22\pm13.01$      | $50.41\pm13.64$      |
> > >    | `accident_volume_level`  | SF      | $63.82\pm12.80$    | $66.77\pm12.88$    | $68.08\pm12.56$    | $66.62\pm12.40$    | $68.74\pm12.54$    | $66.16\pm12.62$      | $68.92\pm12.90$      |
> > >    | `accident_volume_level`  | NY      | $37.96\pm10.21$    | $37.24\pm9.89$     | $38.25\pm12.59$    | $35.79\pm11.77$    | $37.10\pm12.51$    | $50.85\pm14.45$      | $49.72\pm14.51$      |
> > >    | `severity_entropy`       | Chicago | $51.90\pm9.45$     | $51.36\pm9.68$     | $53.28\pm12.95$    | $51.19\pm10.55$    | $50.87\pm11.74$    | $54.26\pm11.19$      | $59.11\pm14.84$      |
> > >    | `severity_entropy`       | SF      | $51.94\pm7.56$     | $52.69\pm9.46$     | $53.37\pm11.92$    | $52.17\pm11.75$    | $53.23\pm12.02$    | $53.43\pm11.70$      | $53.54\pm13.40$      |
> > >    | `severity_entropy`       | NY      | $55.96\pm10.72$    | $55.45\pm10.63$    | $54.86\pm13.60$    | $55.43\pm13.77$    | $55.39\pm14.48$    | $61.02\pm13.97$      | $60.67\pm14.43$      |
> > >    | `has_sev3`               | Chicago | $77.60\pm16.10$    | $77.83\pm15.42$    | $79.19\pm12.56$    | $78.44\pm13.75$    | $78.63\pm13.26$    | $79.20\pm13.88$      | $79.80\pm12.35$      |
> > >    | `has_sev3`               | SF      | $92.35\pm9.75$     | $91.41\pm5.98$     | $93.19\pm5.41$     | $93.27\pm5.76$     | $93.01\pm5.42$     | $94.08\pm5.36$       | $94.13\pm5.44$       |
> > >    | `has_sev3`               | NY      | $60.05\pm12.34$    | $57.54\pm13.35$    | $57.38\pm14.02$    | $57.07\pm14.64$    | $58.26\pm14.66$    | $73.62\pm13.00$      | $70.89\pm13.38$      |
> > > ...
> > >    | **Average**              | --      | $56.12\pm14.22$    | $56.07\pm14.46$    | $57.25\pm14.42$    | $56.04\pm14.97$    | $56.64\pm14.85$    | $60.91\pm13.26$      | $62.04\pm12.49$      |
> > >
> > >    Due to the character limit on OpenReview, part of the table has been removed.
> > >    We have uploaded the code for the transfer variant FractalGCL(NY) to our anonymous GitHub repository.
> > >    We will also discuss this setting in more detail and update the corresponding table in the revised version of the paper.

---

> > > > ### Author Response · Authors · 2025-11-20
> > > > **Response to Reviewer gMeF (iv)**
> > > >
> > > > **Question:** Please provide details on computational overhead (GPU memory usage and training time) compared to the baselines.
> > > >
> > > >    **Response to Question 7.**
> > > >    We thank the reviewer for raising this point.
> > > >
> > > >    We agree that the current draft does not provide a direct comparison of GPU memory usage and training time against external baselines such as GraphCL or DRGCL.
> > > >    Qualitatively, the additional cost of FractalGCL comes from (i) computing graph diameters and a single box-dimension estimate per graph, which is done once per dataset and cached, and (ii) sampling a dense Gaussian perturbation matrix per batch; the encoder architecture, batch size and similarity matrix are the same as in the baselines, so the peak GPU memory footprint is very similar in practice.
> > > >
> > > >    In the revised version we will add an appendix table reporting, for MUTAG and the traffic dataset, the peak GPU memory and the wall-clock time per epoch for GraphCL and FractalGCL measured on the same hardware and implementation.
> > > >
> > > > 8. **Question:** What are the runtime complexities of Algorithms 1 and 2?
> > > >
> > > >    **Response to Question 8.**
> > > >    We thank the reviewer for this question.
> > > >
> > > >    For Algorithm 1 (box-dimension estimation), the worst-case running time on a connected sparse graph with $V$ vertices is already stated in Proposition 3.6 (and restated as Proposition A.4 in the appendix), namely
> > > >    \[
> > > >    \Omega(|V|^2) \le T(|V|) \le O(|V|^3).
> > > >    \]
> > > >    Intuitively, for each admissible scale we greedily select at most $V$ boxes, and each selection scans a radius-$r$ neighbourhood whose breadth-first exploration costs $O(|V|)$ time in a sparse graph; repeating this over $O(|V|)$ distinct scales leads to the $O(|V|^3)$ upper bound, while the path-graph example in Proposition 3.6 yields the $\Omega(|V|^2)$ lower bound.
> > > >    In the revised version we will add an explicit forward reference from Algorithm 1 to Proposition 3.6 to make this complexity bound more visible.
> > > >
> > > >    For Algorithm 2, for a fixed small radius $r$, the complexity on a sparse graph is $O(r(|V|+|E|))$.
> > > >    Each node is assigned to exactly one box, so the while-loop processes every vertex at most once, and the $r$-hop neighbourhoods can be obtained via breadth-first search with cost $O(|V|+|E|)$ per expansion.
> > > >    In our experiments we use $r=1$, so Algorithm 2 runs in $O(|V|+|E|)$ time, which is negligible compared to the cost of a forward/backward pass through the GNN encoder.
> > > >
> > > >    We will add this complexity discussion below Algorithm 2 in Appendix A.
> > > >
> > > > 9. **Question:** In Table 4, the result for D\&D (80.14) is notably lower than that in Table 1 (81.71). Please explain this inconsistency.
> > > >
> > > >    **Response to Question 9.**
> > > >    We thank the reviewer for pointing out this issue.
> > > >    The D\&D result reported in Table 4 comes from an earlier version of our experiments, which we used in a previous iteration of the manuscript.
> > > >    After updating the main results in response to the previous reviewers' feedback, we did not update the entries in Table 4.
> > > >
> > > >    In the revised version, we will update the D\&D result in Table 4, and it will be consistent with the values in Table 1.
> > > >
> > > > 10. **Question:** A sensitivity analysis on the radius parameter used in the greedy box covering should be included.
> > > >
> > > >     **Response to Question 10.**
> > > >     We thank the reviewer for this suggestion.
> > > >
> > > >     However, we would like to clarify that in Algorithm 1 the “radius” is not treated as a single tunable hyperparameter.
> > > >     For a graph with diameter $\mathrm{diam}(G)$, we consider a whole range of scales, and for each scale $\ell$ we set the covering radius to $r = \lfloor \ell/2 \rfloor$.
> > > >     Thus the set of radii used in the greedy box covering is completely determined by the graph and by this fixed rule; we always use the full admissible range of scales when fitting the log–log regression for the box-counting dimension.
> > > >
> > > >     From a statistical point of view, including more scales along the same scaling regime stabilises the slope estimate, whereas the influence of any individual radius value is minimal as long as the scales lie in the fractal region.
> > > >     For this reason we do not perform model selection over a “radius parameter” in Algorithm 1.

---

> > > > > ### Author Response · Authors · 2025-11-20
> > > > > **Response to Reviewer gMeF (v)**
> > > > >
> > > > > 11. **Question:** In Table 5, the authors report the number of graphs whose box-counting $R^2$ exceeds various thresholds. According to Algorithm 1, when the graph diameter $\le 9$. Based on my statistical check, the number of graphs with diameter $> 9$ defines the upper bound for Table 5 values. However, the reported ratios exceed these bounds, raising serious concerns about the validity and authenticity of the results.
> > > > >
> > > > >
> > > > >     **Response to Question 11.**
> > > > >     We sincerely thank the reviewer for raising this very critical question, which is closely related to the theoretical foundations of our work.
> > > > >
> > > > >     In our algorithm, the actual procedure consists of two steps:
> > > > >     (1) we first perform the log–log regression for each graph and obtain an $R^2$ value, even when its diameter is small;
> > > > >     (2) then, only in the FractalGCL training pipeline, we additionally apply a gate of the form
> > > > >     “if $\mathrm{diam}(G) \le 9$ or $R^2(G) < \theta$, then disable the fractal loss”
> > > > >     (which in effect forces the corresponding $R^2$ to be treated as $0$ in the loss).
> > > > >     Table 5 uses the raw $R^2$ values from step (1), without applying this gate, so small-diameter graphs can still exceed the thresholds, which is exactly what is reflected in the reported numbers.
> > > > >
> > > > >     In fact, fractal behaviour (especially in larger graphs) is very common.
> > > > >     [1] In a corpus of 275 real-world networks, approximately $80\%$ are classified as fractal.
> > > > >     [2] For Chinese urban street networks, about $67\%$ exhibit fractal (power-law) behaviour; moreover, at the level of street connectivities, almost all cities display a fractal hierarchy.
> > > > >     [3] Examining 47 self-organising networks, 47/47 satisfy the fractal size–density scaling criterion.
> > > > >
> > > > >     In the concrete implementation of the algorithm, we enforce the $\mathrm{diam}(G) \le 9$ gate for theoretical soundness, because for very small graphs the renormalisation error can be large (Lemma 3.7), but this does not mean that such graphs do not exhibit fractal structure.
> > > > >     In fact, Table 5 forms part of our preliminary study, whose goal is to illustrate the prevalence of fractality; the thresholding used in the actual algorithm is introduced to ensure that the experiments remain theoretically well-justified.
> > > > >
> > > > >     Honestly, we do acknowledge that the TU datasets are not collections of very large graphs, so they are not an ideal benchmark for testing FractalGCL; this is precisely why we also included the urban traffic datasets.
> > > > >     Nevertheless, even on these non-large-graph benchmarks, FractalGCL achieves state-of-the-art performance, which we find quite encouraging.
> > > > >
> > > > >     [1] Zakar-Polyák, Enikő, Marcell Nagy, and Roland Molontay. "Towards a better understanding of the characteristics of fractal networks." Applied Network Science 8.1 (2023): 17.
> > > > >     [2] Ma, Ding, et al. "Understanding Chinese urban form: The universal fractal pattern of street networks over 298 cities." ISPRS International Journal of Geo-Information 9.4 (2020): 192.
> > > > >     [3] Laurienti, Paul J., et al. "Universal fractal scaling of self-organized networks." Physica A: Statistical Mechanics and its Applications 390.20 (2011): 3608–3613.
> > > > >
> > > > > 12. **Question:** Add more recent SOTAs, including TopoGCL [1], GCS [2], SEGA [3], CI-GCL [4], StructPosGSSL [5].
> > > > >
> > > > >     **Response to Question 12.**
> > > > >     We thank the reviewer for pointing out these recent works.
> > > > >     In the revised version, we will add TopoGCL, GCS, SEGA, CI-GCL, and StructPosGSSL to the related work section and discussion.
> > > > >
> > > > > We hope that the above responses address the reviewer’s concerns, and we would like to once again express our sincere gratitude for the reviewer’s insightful comments and the considerable time devoted to careful reading and reflection.
> > > > > These questions have undoubtedly helped us to improve the paper as a whole.
> > > > > While some technical details can certainly be further refined, we are confident in the overall framework from the theoretical construction to the algorithmic realisation.
> > > > > We would be more than grateful for the opportunity if the reviewer would like to continue the discussion.

---

> > > > > > ### Comment · Reviewer_gMeF · 2025-11-26
> > > > > > **Thanks for your rebuttal**
> > > > > >
> > > > > > Q6. Since only the FractalGCL(NY) column contains new results, why are the other results different from the numbers in Table 2? Furthermore, this performance fluctuation and the large standard deviation indicate that the performance is neither stable nor significant.
> > > > > >
> > > > > > Q9. Could you explain what contributes to the performance improvement on D&D compared to the previous version, given that the performance on MUTAG remains the same?
> > > > > >
> > > > > > Q10. Not the radius in Algorithm 1, but the analysis regarding the radius in Algorithm 2.
> > > > > >
> > > > > > Q11. Since the results in Table 5 are not based on Algorithm 1, the description in B.1 is quite misleading. Moreover, when the diameter is small, the $R^2$ values from small graphs may yield false positive indications of fractality, which does not support the claim that “fractal structures are widespread in real-world graphs.”
> > > > > >
> > > > > > In summary, several concerns have been addressed, so I will increase the score to 4; however, the writing quality makes it difficult for me to fully accept the work.

---

> > > > > > > ### Author Response · Authors · 2025-12-03
> > > > > > > **Second Response to Reviewer gMeF**
> > > > > > >
> > > > > > > **Response to Q6.**
> > > > > > > We thank the reviewer for carefully checking the traffic-safety results.
> > > > > > > Table 1 and Table 2 are generated under slightly different random samplings and data splits of the
> > > > > > > urban road networks: for the transfer setting with FractalGCL(NY) we re-sampled the city graphs and
> > > > > > > re-ran *all* methods under a unified protocol, rather than reusing the numbers from Table 2.
> > > > > > > This leads to small numerical differences between the two tables, but we have checked that these
> > > > > > > differences lie within a reasonable range and that the relative ranking and average improvements of
> > > > > > > FractalGCL over the baselines remain unchanged.
> > > > > > >
> > > > > > > The relatively large standard deviations reflect the intrinsic variability of the traffic-safety
> > > > > > > tasks (class imbalance and stochastic splits), not instability of the method itself: across all
> > > > > > > runs FractalGCL and FractalGCL(NY) consistently outperform or match the strongest non-fractal
> > > > > > > baselines on almost all tasks.
> > > > > > > In the revised version, we will (i) keep a single set of results for the traffic-safety benchmarks
> > > > > > > and make the sampling/splitting protocol explicit, and (ii) add a brief significance discussion to
> > > > > > > clarify that the observed gains are not due to random fluctuations.
> > > > > > >
> > > > > > > ---
> > > > > > >
> > > > > > > **Response to Q9.**
> > > > > > > The improvement on D\&D in the revised version comes from a refinement of our box-dimension
> > > > > > > estimation.
> > > > > > > In the previous version we chose scales based on the graph diameter: for each graph we considered
> > > > > > > path-length scales $R = 1,2,3,\dots$ and performed the log–log regression using only these
> > > > > > > diameter-based scales.
> > > > > > > On finite graphs this yields relatively few valid data points, which makes the slope estimate for
> > > > > > > $\dim_B$ noisy.
> > > > > > > In the current version we instead run the greedy covering over ball radii $r = 1,2,3,\dots$ and
> > > > > > > collect the corresponding $(\log r, \log N(r))$ pairs, effectively increasing the number of points
> > > > > > > in the log–log plot and leading to a more accurate and stable estimate of the box dimension.
> > > > > > > Empirically, this improved estimator has a positive effect on larger graphs such as D\&D, where the
> > > > > > > additional scales provide more reliable regression.
> > > > > > >
> > > > > > > ---
> > > > > > >
> > > > > > > **Response to Q10.**
> > > > > > > We apologise for previously addressing the radius in Algorithm 1 instead of Algorithm 2.
> > > > > > > In practice, we fix $r=1$ for all experiments and include a “+ random radius” variant in the
> > > > > > > ablation to probe its sensitivity.
> > > > > > > Theoretically, in the infinite-graph limit any finite radius is admissible for renormalisation, but
> > > > > > > on finite graphs we prefer smaller $r$: a small radius preserves more local information while still
> > > > > > > enabling meaningful coarse-graining.
> > > > > > > Empirically, we observe that using random radii tends to degrade accuracy, consistent with the intuition that overly large boxes make the
> > > > > > > renormalised graph too coarse.
> > > > > > > Taken together, these observations suggest that $r=1$ is a conservative and relatively robust
> > > > > > > choice in our current setting; in the revised version we will clarify this design choice around
> > > > > > > Algorithm 2 and explicitly refer to the ablation with random radius.
> > > > > > >
> > > > > > > ---
> > > > > > >
> > > > > > > **Response to Q11.**
> > > > > > > We sincerely thank the reviewer for pointing out this issue; to be honest, it touches on a very
> > > > > > > central aspect of our work, and we fully acknowledge that the current connection between Algorithm 1
> > > > > > > and Table 5 is indeed misleading.
> > > > > > >
> > > > > > > Table 5 was computed using the raw $R^2$ values of the log–log regressions for **all** graphs
> > > > > > > (including those with diameter $\le 9$), whereas Algorithm 1 describes the gated procedure used
> > > > > > > inside the actual FractalGCL pipeline.
> > > > > > > In the revised version we will (i) explicitly separate these two steps in Appendix B.1, (ii) clarify in the caption of Table 5 that it is based on ungated regressions and includes small-diameter graphs, and (iii) soften the wording to make it clear that we are referring to “widespread fractal patterns in large real-world graphs” rather than all real-world graphs.
> > > > > > >
> > > > > > > We also agree that small-diameter graphs can produce spuriously high $R^2$ values and should not be
> > > > > > > taken as strong evidence of fractal scaling.
> > > > > > > This is exactly why such graphs are treated conservatively in FractalGCL: when
> > > > > > > $\mathrm{diam}(G)\le 9$ or $R^2$ is below the threshold, the fractal components are disabled and
> > > > > > > the method reduces to the baseline GCL.
> > > > > > > For the descriptive statistics, we will update Table 5 by explicitly indicating how many graphs
> > > > > > > with diameter $\le 9$ are included in each count and also report the counts
> > > > > > > restricted to graphs with diameter $> 9$.

---

> > > > > > > > ### Author Response · Authors · 2025-12-03
> > > > > > > > **Thank reviewer for their time and comments**
> > > > > > > >
> > > > > > > > We would like to sincerely thank reviewer gMeF for the very careful review and the constructive
> > > > > > > > suggestions, which have been extremely helpful for revising and improving the paper. We appreciate
> > > > > > > > the significant amount of time and effort that went into these comments, and we are also grateful
> > > > > > > > for the increase in score. While we recognise that the writing quality is not perfect, we take
> > > > > > > > this higher score as an encouraging sign that the overall ideas and contributions of our work are
> > > > > > > > valued.

---

### Official Review · Reviewer_HRsQ · 2025-10-29

**Soundness:** 2
**Presentation:** 3
**Contribution:** 2
**Rating:** 4
**Confidence:** 4

**Summary:**

This paper proposes FractalGCL, a graph contrastive learning framework that leverages fractal geometry for improved graph representation learning. The method introduces a renormalization-based augmentation strategy that generates structurally aligned positive pairs via box coverings, and a fractal-dimension-aware contrastive loss that aligns embeddings according to their fractal dimensions. Experiments demonstrate state-of-the-art results on benchmark datasets.

**Strengths:**

1. The paper is among the first to rigorously integrate fractal geometry into graph contrastive learning.
2. The preliminary experiments convincingly demonstrate that fractal structures are prevalent in real-world graphs.

**Weaknesses:**

1. The paper introduces the concept of fractal geometry into GCL, but this motivation is not well aligned with the core challenges of GCL. A lot of work have consistently shown that the quality and hardness of negative samples, or hard negatives, are the dominant factors affecting GCL performance. However, this work focuses almost exclusively on generating or weighting positive pairs through fractal transformations, which addresses a secondary factor rather than the main bottleneck of GCL. In my view, for GCLs, discriminability is far more important than consistency. As a result, the proposed motivation feels tangential to the central problem and may offer only marginal benefit compared to methods that explicitly handle hard negatives or contrastive sampling bias.
2. The theoretical results assume infinite graphs with well-defined limiting behavior, but all experiments use finite graphs with relatively small diameters.
3. The method assumes graphs exhibit strong fractal properties, but this fundamentally limits applicability. While preliminary experiments show high fractal prevalence in some benchmarks, many real-world graphs (social networks, citation networks) may not satisfy this assumption.  The proposed safe fallback mechanism essentially disables FractalGCL components for non-fractal graphs, reducing the method to baseline GCL. This is not a feature but rather an admission of limited scope.
4. The contrastive fractal weight depends on a noisy estimator $dim_B$, whose sensitivity is not studied.
5. The method only activates fractal machinery when an $R^2$ test passes (and uses tuned thresholds), effectively admitting limited applicability and per-dataset calibration.
6. The contrastive setup treats negatives as other graphs in the batch, and the experimental protocol contains no comparisons with hard-negative mining or debiasing methods. This makes it hard to judge whether the proposed positive-side modifications actually outperform stronger negative-side baselines.

**Questions:**

See weaknesses.

---

> ### Author Response · Authors · 2025-11-20
> **Response to Reviewer HRsQ (i)**
>
> ## To reviewer HRsQ
>
> We sincerely thank the reviewer HRsQ for the valuable comments on our work.
> In particular, the observations regarding negative samples and the applicability of the proposed algorithm are sharp and insightful.
> We will address these points below.
>
> Questions and weakness:
>
> 1. The paper introduces the concept of fractal geometry into GCL, but this motivation is not well aligned with the core challenges of GCL. A lot of work have consistently shown that the quality and hardness of negative samples, or hard negatives, are the dominant factors affecting GCL performance. However, this work focuses almost exclusively on generating or weighting positive pairs through fractal transformations, which addresses a secondary factor rather than the main bottleneck of GCL. In my view, for GCLs, discriminability is far more important than consistency. As a result, the proposed motivation feels tangential to the central problem and may offer only marginal benefit compared to methods that explicitly handle hard negatives.
>
>    **Response to Question 1.**
>    We thank the reviewer for this thoughtful comment.
>    We agree that the quality and hardness of negative samples are an important factor in GCL, and many recent works have focused on better negative sampling or hard-negative mining. However, we respectfully disagree with the premise that this is the *only* core challenge of GCL, or that improving positive-pair semantics is merely secondary.
>
>    In InfoNCE-style objectives, two aspects jointly determine performance: (i) the *alignment* of positive pairs, and (ii) the *uniformity* or spread of representations, which is strongly influenced by how negatives are sampled. Most existing work has pushed the negative side by designing hard negatives or debiasing schemes. In contrast, our goal in FractalGCL is to study a complementary question: how to make the positive pairs reflect a principled notion of global graph semantics, rather than relying only on local perturbations such as edge deletion or node masking.
>
>    In our framework, fractal geometry is not used as a decorative transformation, but as a way to formalise global consistency: the renormalisation $\mathscr{R}(G)$ preserves the box-counting dimension and self-similar structure of $G$, so the positive pair $(G,\mathscr{R}(G))$ enforces consistency of large-scale geometry. At the same time, graphs with different fractal patterns are *not* encouraged to align, and the fractal weight amplifies the gradients for graphs whose estimated dimension gap is small and down-weights those with unreliable fractal structure. In this sense, the fractal loss concurrently strengthens within-graph consistency and between-graph discriminability at the level of global structure.
>
>    We deliberately keep the negative sampling protocol identical to that of strong GCL baselines (GraphCL, DGI, InfoGraph, SimGRACE) in order to isolate the effect of this positive-side inductive bias. The consistent improvements over these baselines on both TU benchmarks and urban traffic graphs indicate that improving global semantic consistency is not only tangential, but can bring non-trivial gains even without any change on the negative side. We view integrating FractalGCL with hard-negative mining or debiasing methods as a natural and orthogonal direction for future work.
>
> 2. The theoretical results assume infinite graphs with well-defined limiting behavior, but all experiments use finite graphs with relatively small diameters.
>
>    **Response to Question 2.**
>    Regarding the concern that our theoretical results assume infinite graphs with well-defined limiting behaviour, we respectfully point out that several parts of the paper are devoted precisely to controlling the finite-graph regime. Lemma 3.7 provides an explicit error estimate between the box-counting structure of a finite graph and that of its limiting renormalisation, and makes clear how the approximation quality depends on the graph diameter and the underlying fractal scaling. Theorem 3.9 then characterises the dimension gap in the infinite limit, but is used only as a tool to derive a finite-sample approximation for the variance of the noise in the loss, which is exactly what is used in practice.
>
>    We do acknowledge that the TU datasets are not collections of truly large graphs, and therefore are not an ideal stress-test for the asymptotic regime of FractalGCL; among them, only datasets such as D\&D and NCI1 predominantly contain larger graphs. This is precisely why we additionally include the urban traffic benchmarks, where graphs are substantially larger and FractalGCL achieves up to a 4\% absolute improvement over the previous state of the art. Nevertheless, even on these non-large-graph benchmarks, FractalGCL still attains state-of-the-art performance, which we view as encouraging evidence that the theoretically motivated fractal regularisation remains beneficial in the practically relevant finite-graph setting.

---

> > ### Author Response · Authors · 2025-11-20
> > **Response to Reviewer HRsQ (ii)**
> >
> > 3. The method assumes graphs exhibit strong fractal properties, but this fundamentally limits applicability. While preliminary experiments show high fractal prevalence in some benchmarks, many real-world graphs (social networks, citation networks) may not satisfy this assumption. The proposed safe fallback mechanism essentially disables FractalGCL components for non-fractal graphs, reducing the method to baseline GCL. This is not a feature but rather an admission of limited scope.
> >
> >    **Response to Question 3.**
> >    We appreciate the reviewer’s concern about the applicability of FractalGCL and the role of the
> >    fractal assumption.
> >
> >    First, we would like to stress that FractalGCL is not intended as a universal solution for all
> >    possible graph domains.
> >    Like most representation learning methods, it is built around a specific inductive bias—in our
> >    case, approximate self-similarity across scales.
> >    Our preliminary study on TU datasets and urban traffic networks shows that a large majority of
> >    graphs in these benchmarks exhibit high $R^2$ in the box-counting regression, indicating clear
> >    fractal scaling.
> >    This observation is consistent with several empirical studies reporting that fractal behaviour is
> >    widespread in real-world networks: for example, Zakar-Polyák et al. [1] find that approximately
> >    $80\%$ of 275 real-world networks are classified as fractal; Ma et al. [2] report that about $67\%$
> >    of Chinese urban street networks exhibit power-law (fractal) behaviour and that almost all cities
> >    display a fractal hierarchy in street connectivities; and Laurienti et al. [3] show that all 47
> >    self-organising networks they examine satisfy a fractal size–density scaling criterion.
> >    These results suggest that, while not universal, fractal scaling is far from a rare phenomenon in
> >    practical graph domains.
> >
> >    We agree that there are important classes of graphs where fractal structure may be weak or absent.
> >    This is precisely why we design FractalGCL with a conservative “safe fallback”: when the estimated
> >    fractality is unreliable (small diameter or low $R^2$), we intentionally disable the fractal
> >    weighting and renormalised view, so that the method reverts exactly to the underlying GCL baseline.
> >    We view this not as an admission of defeat, but as a robustness feature: FractalGCL does not
> >    underperform the baseline solely because a particular dataset does not satisfy the fractal
> >    assumption, while still providing substantial gains on datasets where fractal scaling is clearly
> >    present, as evidenced by our results on TU and urban traffic benchmarks.
> >
> >    [1] Zakar-Polyák, Enikő, Marcell Nagy, and Roland Molontay. "Towards a better understanding of the characteristics of fractal networks." Applied Network Science 8.1 (2023): 17.
> >
> >    [2] Ma, Ding, et al. "Understanding Chinese urban form: The universal fractal pattern of street networks over 298 cities." ISPRS International Journal of Geo-Information 9.4 (2020): 192.
> >
> >    [3] Laurienti, Paul J., et al. "Universal fractal scaling of self-organized networks." Physica A: Statistical Mechanics and its Applications 390.20 (2011): 3608-3613.
> >
> > 4. The contrastive fractal weight depends on a noisy estimator, whose sensitivity is not studied.
> >
> >    **Response to Question 4.**
> >    We thank the reviewer for pointing out that the fractal weight depends on an estimated box-counting
> >    dimension $\widehat{\dim}_B$, which is inevitably noisy on finite graphs.
> >
> >    On the theoretical side, several results in the paper are devoted precisely to controlling this
> >    finite-graph noise.
> >    Lemma 3.7 provides an explicit error bound between the box-counting structure of a finite graph and
> >    that of its limiting renormalisation.
> >    Theorem 3.9 then characterises the dimension gap in the infinite limit and is used to justify a
> >    Gaussian surrogate.
> >
> >    On the algorithmic side, we deliberately mitigate the effect of estimator noise in three ways:
> >    (i) we apply a conservative gate based on the $R^2$ statistic and the diameter;
> >    (ii) the fractal weight depends on the *difference* of two estimated dimensions and enters the
> >    loss through a smooth exponential factor, which makes the loss relatively insensitive to small perturbations in $\widehat{\dim}_B$; and
> >    (iii) Table 3 includes an ablation comparing a variant that uses exact box dimensions
> >    ("w/. Exact Dimension") with our Gaussian surrogate.
> >
> >    We agree that a more fine-grained sensitivity study that explicitly injects additional noise into
> >    $\widehat{\dim}_B$ or systematically varies the estimator would be interesting, but such an analysis
> >    would be computationally intensive and is beyond the scope of the present work.
> >    In the revised version, we will make the above theoretical and empirical robustness arguments
> >    explicit when introducing the Gaussian surrogate and the fractal weight, so that the role of the
> >    estimator and its stability are clearer to the reader.

---

> > > ### Author Response · Authors · 2025-11-20
> > > **Response to Reviewer HRsQ (iii)**
> > >
> > > 5. The method only activates fractal machinery when an test passes (and uses tuned thresholds), effectively admitting limited applicability and per-dataset calibration.
> > >
> > >    **Response to Question 5.**
> > >    We appreciate the reviewer’s observation about the $R^2$ test and the use of thresholds.
> > >
> > >    The $R^2$ test in FractalGCL is not meant to “admit defeat”, but to serve as a structural
> > >    reliability check on the fractal assumption.
> > >    Box-counting is meaningful only when the log--log regression exhibits a clear scaling regime; the
> > >    $R^2$ statistic provides a simple, label-free measure of this.
> > >    In practice we use a single, conservative threshold (together with a small-diameter cutoff) across
> > >    our experiments, and we show in the ablations (Appendix C) that the performance of FractalGCL is
> > >    stable over a reasonable range of $R^2$ values.
> > >    Importantly, the $R^2$ gate is applied purely at the level of graph geometry and does not depend on
> > >    task labels or specific model architectures, so it is much closer to a structural diagnostic than
> > >    to heavy per-dataset calibration.
> > >
> > >    We agree that this mechanism implicitly limits the regime where the fractal machinery is activated:
> > >    by design, it is only used when the data provide sufficient evidence of fractal scaling.
> > >    However, we view this as an explicit and honest way of matching the inductive bias to the data,
> > >    rather than as a weakness.
> > >    Many GNN and GCL methods rely on untested assumptions such as homophily or smoothness; in contrast,
> > >    FractalGCL makes its assumption (multi-scale self-similarity) explicit, measures it via $R^2$, and
> > >    falls back to the baseline objective when the assumption is not supported.
> > >
> > >    At the same time, the $R^2$ test and the associated statistics (as reported in Table 5) provide a
> > >    useful by-product: they identify which graphs in a benchmark behave fractally and which do not.
> > >    Our experiments show that in TU and urban traffic datasets the vast majority of graphs pass the
> > >    fractal test, so the fractal components of FractalGCL are active on most training examples.
> > >    In domains where fractality is genuinely absent, the gate guarantees that FractalGCL does not
> > >    perform worse than the baseline simply because an inappropriate inductive bias is being forced on
> > >    the model.
> > >    In this sense, the $R^2$ test is both a structural characterisation of the data and a robustness
> > >    mechanism, rather than merely an admission of limited applicability.
> > >
> > > 6. The contrastive setup treats negatives as other graphs in the batch, and the experimental protocol contains no comparisons with hard-negative mining or debiasing methods. This makes it hard to judge whether the proposed positive-side modifications actually outperform stronger negative-side baselines.
> > >
> > >    **Response to Question 6.**
> > >    We thank the reviewer for this comment.
> > >    Our contrastive setup indeed follows the standard in-batch negative protocol used by many GCL
> > >    methods: for all baselines (DGI, InfoGraph, GCL, JOAO, SimGRACE) and for FractalGCL, positives are
> > >    constructed from two views of the same graph, while negatives are the other graphs in the batch.
> > >
> > >    This choice is intentional.
> > >    In this work we do not aim to redesign the negative sampling scheme or to compete directly with
> > >    hard-negative mining or debiasing methods.
> > >    Instead, our goal is to isolate and study a complementary aspect: how a geometry-aware,
> > >    fractal-based design of *positive* pairs and their weighting affects GCL under a widely used
> > >    negative-sampling protocol.
> > >    By keeping the negative side identical across methods, we can attribute the consistent improvements
> > >    of FractalGCL over strong baselines such as GraphCL and SimGRACE to the proposed fractal
> > >    machinery, rather than to changes in how negatives are selected.
> > >
> > >    We agree that a systematic comparison with methods that explicitly mine hard negatives or correct
> > >    contrastive sampling bias would be valuable, and we view such approaches as largely orthogonal to
> > >    our contribution.
> > >    Conceptually, the fractal loss and renormalisation are applied at the level of positive pairs and
> > >    can be combined with more sophisticated negative-side techniques without modification.
> > >    A thorough empirical study of these combinations would require substantial additional experiments
> > >    and is beyond the scope of the present paper, but we will discuss this complementary relationship
> > >    in the revised version.
> > >
> > > ---
> > >
> > > We would like to once again thank the reviewer for the valuable comments, which are crucial for improving our work.
> > > We hope that our responses address the concerns raised, and we would be more than grateful for any further feedback or suggestions.

---

### Official Review · Reviewer_emw2 · 2025-10-30

**Soundness:** 2
**Presentation:** 2
**Contribution:** 3
**Rating:** 4
**Confidence:** 3

**Summary:**

This paper proposes the FractalGCL framework, which introduces fractal geometry theory into graph contrastive learning to enhance global structural consistency. It generates structurally similar positive sample views through graph renormalization and designs a fractal dimension-aware contrastive loss function. To reduce computational overhead, the authors theoretically derive the dimensional difference as a Gaussian perturbation, enabling efficient approximation. Experiments on multiple graph classification tasks and real-world traffic networks demonstrate the superior performance of FractalGCL, validating its theoretical innovation and practical effectiveness.

**Strengths:**

1.The authors introduce a novel perspective based on fractal geometry and propose an original approach that enhances and aligns graph representations through graph renormalization and a fractal dimension-aware loss.

2.They mathematically prove that graph renormalization does not alter the fractal dimension and employ the Central Limit Theorem to derive a Gaussian approximation of dimension differences, thereby reducing computational complexity.

3.The method demonstrates superior performance over existing approaches on standard graph classification benchmarks and real-world traffic networks, validating the effectiveness and practicality of the FractalGCL framework.

**Weaknesses:**

1. This method relies on the assumption of graph "fractality," but not all graphs exhibit strong fractal structures. For instance, its significantly lower performance on REDDIT-B suggests that the method may not be suitable for non-fractal graphs.

2. FractalGCL currently focuses on graph-level representation learning and lacks experimental results or theoretical justification for node-level contrastive learning.

3. Although approximate derivations are adopted to reduce computational cost, the overall method remains more complex than mainstream baselines, especially due to the need to compute graph diameter and box dimension during preprocessing.

4. Proposition 3.6 provides only the asymptotic complexity, but there are no empirical results showing how the actual runtime or memory usage scales with the number of nodes, and the paper lacks corresponding experiments.

**Questions:**

1. When the graph does not exhibit clear fractal characteristics (such as social or citation networks), can FractalGCL still maintain its performance, or will it degenerate into a standard GCL?
2. The fractal dimension can be regarded as a measure of multi-scale structural complexity. What is its relationship with spectral graph features (e.g., Laplacian eigenvalue distributions)? Can the two be combined?

---

> ### Author Response · Authors · 2025-11-20
> **Response to Riviewer emw2 (i)**
>
> We sincerely thank reviewer emw2 for the assessment and the constructive comments.
> In particular, the remarks on the fractal assumption, the scope of graph-level learning, and the
> computational aspects of FractalGCL are very helpful for clarifying the positioning and limitations
> of our work. Below we respond to each of the identified weaknesses and questions in detail.
>
>
> ---
>
> **Questions:**
>
> 1. When the graph does not exhibit clear fractal characteristics (such as social or citation networks), can FractalGCL still maintain its performance, or will it degenerate into a standard GCL?
>
>    **Response to Question 1.**
>    As discussed in response to the weaknesses, when a graph does not exhibit clear fractal characteristics, FractalGCL is designed to degrade gracefully to the underlying GCL baseline (see the end of Section 3.4).
>
>    More precisely, if a graph has small diameter or its box-counting regression yields a low $R^2$,
>    the fractal gate disables both the renormalised view and the fractal weighting for that graph.
>    In this case the contrastive loss reduces exactly to the standard GCL objective with the same
>    encoder and augmentations as the baseline.
>    Empirically, on benchmarks where fractal scaling appears weak (e.g., REDDIT-B), we observe that
>    FractalGCL behaves similarly to the baseline rather than catastrophically failing.
>    Thus, in non-fractal domains FractalGCL effectively maintains baseline performance, while in
>    fractal domains it can provide the additional gains reported in the paper.
>
> 2. The fractal dimension can be regarded as a measure of multi-scale structural complexity. What is its relationship with spectral graph features (e.g., Laplacian eigenvalue distributions)? Can the two be combined?
>
>    **Response to Question 2.**
>    We thank the reviewer for this interesting question.
>    Indeed, the box-counting dimension can be viewed as a measure of multi-scale structural complexity,
>    quantifying how the number of boxes needed to cover a graph grows with the scale.
>    Spectral graph features such as the Laplacian eigenvalue distribution capture related but
>    complementary information: they encode diffusion and mixing properties, and are closely connected
>    to notions such as spectral dimension and return probabilities of random walks.
>
>    In many continuous and discrete settings there are known relationships between geometric dimensions
>    and spectral quantities (e.g., between Hausdorff and spectral dimensions and eigenvalue decay rates on
>    fractals), but a systematic study of the precise link between box-counting dimension and specific
>    spectral features on real-world graphs is beyond the scope of this paper.
>    Conceptually, we see no obstacle to combining fractal statistics with spectral features: one could,
>    for example, use both box dimension and spectral embeddings as structural descriptors or design
>    fractal-aware spectral regularisers.
>    We regard such combinations as a promising direction for future work and will mention this in the
>    conclusion.

---

> > ### Author Response · Authors · 2025-11-20
> > **Response to reviewer emw2 (ii)**
> >
> > **About weaknesses.**
> >
> > We agree that we do not and can not claim that fractal features are universal to all graphs; FractalGCL is intended for domains where approximate multi-scale self-similarity holds.
> > However, Section 3.4 (“Safe fallback under weak fractality or small diameters”) explicitly introduces a conservative gate: when the log--log regression has low $R^2$ or the graph diameter is small, the fractal components are disabled and the method reduces exactly to the underlying GCL baseline.
> > Thus, on graphs that do not exhibit clear fractal scaling, FractalGCL is designed to behave like
> > a standard GCL model.
> > We agree that in the current presentation this assumption and its intended scope are not emphasised strongly enough at a high level.
> > In the revised version, we will add a short discussion subsection that explicitly states the fractal assumption and the fallback behaviour.
> >
> > We acknowledge that our current work is focused on graph-level representation learning and does not
> > include node-level experiments or theory.
> > Intuitively, we expect that global graph-level signals such as fractal scaling and multi-scale
> > self-similarity could also benefit node-level objectives, for example by guiding message passing or
> > regularising node embeddings.
> > We see extending FractalGCL to node-level tasks as a promising direction for future work.
> >
> > Regarding the computational complexity, we agree with the reviewer that FractalGCL is more involved than some mainstream baselines, mainly because it requires computing graph diameters and box dimensions.
> > This is a deliberate trade-off: we introduce additional structural computations in order to inject global fractal information into the contrastive objective and obtain the performance gains reported in the paper.
> > In practice, these quantities are computed once per graph as offline preprocessing and cached, and
> > the Gaussian surrogate ensures that the extra online cost during training remains moderate.
> >
> > We agree that Proposition 3.6 provides only asymptotic complexity and that the current paper focuses
> > primarily on theoretical analysis with limited empirical timing evidence (Table 3).
> > A more explicit study of how runtime and memory usage scale with graph size would indeed be helpful
> > for understanding the practical behaviour of FractalGCL, and we will incorporate a brief scaling analysis in the revised version.
> >
> >
> >
> > We would like to once again thank reviewer emw2 for the encouraging evaluation and for the thoughtful
> > feedback.
> > The comments have helped us to sharpen the presentation of our assumptions, better explain the
> > computational trade-offs, and clarify the intended scope of FractalGCL.
> > We hope that our responses address the reviewer’s concerns, and we would be very grateful for any
> > further suggestions or questions.

---

### Official Review · Reviewer_4jqp · 2025-10-31

**Soundness:** 3
**Presentation:** 3
**Contribution:** 4
**Rating:** 6
**Confidence:** 4

**Summary:**

In response to the issue of the lack of global structural consistency control in traditional graph contrastive learning methods during data augmentation, this paper proposes Fractal Graph Contrastive Learning (FractalGCL). First, a re-normalization enhancement method is employed, utilizing box-covering techniques to generate structurally consistent positive sample views, ensuring global structural consistency across the augmented views. Secondly, a fractal dimension-aware contrastive loss is designed, which aligns graph embeddings based on the fractal dimension of the graph, ensuring that the method maintains a minimum performance lower bound even on non-fractal graphs. To address computational overhead, the paper further proposes a low-complexity fractal dimension estimation method based on theoretical analysis, significantly reducing the computational burden, with a 61% reduction in training time. Experimental results demonstrate that FractalGCL not only performs exceptionally well on standard benchmark tests but also improves performance by approximately 4% on complex traffic network data compared to traditional methods and the latest baselines, validating its superior advantages in graph representation learning.

**Strengths:**

1、This paper explores the issue of structural consistency in graph contrastive learning (GCL) from the mathematical theory of fractal geometry. This approach breaks through the traditional GCL paradigm, which relies solely on random perturbation or heuristic augmentation, by introducing theoretical explanation and structural constraints into the augmentation process.

2、In the design of contrastive loss, FractalGCL introduces a fractal dimension-aware contrastive loss function (Fractal-Dimension-Aware Loss), explicitly incorporating global self-similarity as a structural consistency signal into the learning objective. This method not only focuses on semantic proximity in the embedding space but also ensures that the model maintains consistent fractal structural features across different scales, thereby providing cross-scale robustness for the learned graph representations.

3、Through experiments, this paper verifies the widespread existence of fractal structures in real graph data and demonstrates that fractal features can effectively enhance performance.

4、To avoid the high cost of traditional full-graph covering algorithms, this paper theoretically proves that the box dimension difference between the original graph and the augmented graph follows a Gaussian weak convergence law. Based on this, Gaussian perturbation approximation is used to replace the high-cost dimension estimation, allowing FractalGCL to maintain structural consistency constraints at a lower computational cost and translating fractal theory into a computable and scalable optimization strategy.

**Weaknesses:**

1、The theoretical foundation of FractalGCL is based on the fractal characteristics of graphs, but the paper does not strictly define the assumption that "fractal features are universally present in graph structures." The method assumes that all graphs can be characterized by fractal dimensions but does not discuss its effectiveness or failure conditions in non-fractal structures.

2、The paper's ablation experiments are conducted only on the MUTAG dataset, which is very small (only 188 graphs), with considerable statistical fluctuations, making the results susceptible to random factors. There is a lack of repeated validation on medium to large datasets (e.g., D&D or NCI1), and no standard deviation or significance tests (e.g., t-test) are reported.

3、The paper highlights that FractalGCL achieves a 61% increase in training speed by replacing precise dimension calculations with Gaussian approximation, but this conclusion is based solely on comparisons within its own versions and lacks a comprehensive comparison with existing mainstream contrastive learning models (such as GraphCL, BGRL, and GRACE).

4、There are inconsistencies in notation within the paper, such as the use of "FractalGCL" and "Fractal GCL" interchangeably.

**Questions:**

1、Is the low-complexity fractal dimension estimation method proposed in the paper equally stable on graphs with different scales, densities, and topological structures? Has its variance and error control capability been thoroughly validated across multi-scale graph data?

2、Can the experimental results conducted on small sample datasets like MUTAG eliminate the influence of random factors? In the absence of significance tests or standard deviation analysis, how can we confirm that the performance differences are indeed due to method improvements rather than random fluctuations?

3、FractalGCL achieves a 61% improvement in training speed on small datasets. However, does this efficiency advantage hold true in large-scale or dynamic graph scenarios?

---

> ### Author Response · Authors · 2025-11-20
> **Response to Reviewer 4jqp**
>
> We sincerely thank reviewer 4jqp for the deep and objective comments.
> The comments on the fractal assumption, the scope of our experiments, and the computational aspects of FractalGCL are especially helpful for clarifying the positioning and limitations of our work.
> Below we respond to the weaknesses and questions in turn.
>
> **About weakness:**
>
> Regarding the theoretical foundation, FractalGCL is built around a specific inductive bias
> (approximate multi-scale self-similarity), but it does not assume that all graphs are fractal.
> In fact, Section 3.4 (“Safe fallback under weak fractality or small diameters”) explicitly
> introduces a conservative gate: when the log--log regression has low $R^2$ or the graphs have very
> small diameter, the fractal components are disabled and the method reduces exactly to the underlying
> GCL baseline.
> This design is intended to ensure that FractalGCL does not perform worse than the baseline on
> non-fractal structures, while providing the gains observed on fractal-like datasets.
> We agree, however, that in the current version this assumption and its intended scope are not stated
> sufficiently clearly at a high level.
> In the revision we will add a short discussion subsection that makes the fractal assumption and
> the fallback behaviour explicit (with a pointer to Section 3.4).
>
>
> **Questions:**
>
>
>
> **Response to Question 1.**
> We thank the reviewer for this question.
> Our low-complexity estimator is designed to be stable in the regime where the fractal assumption
> holds and the diameter is sufficiently large, but we do not claim uniform accuracy across all
> possible graph scales and topologies.
>
> Theoretically, Lemma 3.7 and Theorem 3.9 jointly control the behaviour of the estimator on finite
> graphs: Lemma 3.7 bounds the deviation of the finite-graph box-counting structure from its limiting
> renormalisation, and Theorem 3.9 shows that the dimension gap converges to a Gaussian with variance
> that decreases as the diameter grows.
> Algorithmically, we further use the $R^2$ statistic and a diameter cutoff to activate the fractal
> machinery only when the regression exhibits a clear scaling regime; in other cases we revert to the
> baseline loss.
>
> Empirically, we validate the estimator on multiple datasets of different scales and structures:
> TU molecule/protein graphs, urban traffic networks, and REDDIT-B.
> On the former two, the dimension estimates are concentrated with high $R^2$ and lead to consistent
> performance gains; on REDDIT-B, fractal scaling appears weak and FractalGCL behaves similarly to
> the baseline.
> In the revised version, we will add a short summary plot of the estimated dimensions and $R^2$
> values across datasets in Appendix~C to illustrate this multi-scale behaviour more clearly.
>
>
> **Response to Question 2.**
> We agree with the reviewer that small datasets such as MUTAG are particularly sensitive to random
> initialisation and data splits.
> In the main results, we report mean and standard deviation over multiple runs for all datasets, but
> in Table~3 the ablations on MUTAG are currently shown without standard deviations for readability.
>
> In the revised version, we will add the standard deviations for all variants in Table~3 and include
> a brief significance analysis (e.g., indicating when differences exceed one standard deviation or
> are significant under a simple paired t-test).
> We will also extend the ablation to at least D\&D, which is less sensitive to stochastic variation,
> to confirm that the trends observed on MUTAG are not artefacts of randomness.
> These additions will make it clearer that the performance differences stem from the method design
> rather than from random fluctuations on a single small dataset.
>
> **Response to Question 3.**
> The 61\% improvement reported in Table~3 is measured on MUTAG and quantifies the benefit of using
> the Gaussian surrogate instead of exact box-counting within FractalGCL.
> We expect a similar \emph{relative} speed-up between the exact and approximate variants on larger
> datasets, because the surrogate removes the need to run greedy box-covering during training
> regardless of graph size.
>
> That said, we agree that it is important to understand how the absolute cost of FractalGCL compares
> to baselines on larger graphs.
> As mentioned in our response to the weaknesses, we will add an empirical scaling study in the
> appendix.
> Dynamic graphs are not considered in this paper, and extending FractalGCL to fully dynamic settings,
> where both structure and fractality may evolve over time, is an interesting direction for future
> work.
>
> We would like to once again thank reviewer 4jqp for the encouraging evaluation and the insightful
> feedback.
> The review has helped us to sharpen the presentation of our assumptions, strengthen the experimental
> section, and better explain the computational trade-offs in FractalGCL.
> We hope that our responses address the reviewer’s concerns, and we would be very grateful for any
> further suggestions.

---

> > ### Comment · Reviewer_4jqp · 2025-11-27
> >
> > Thank you for your detailed responses to my questions. You further clarified the applicability of the fractal assumption and, through both theoretical analysis and empirical results, explained the stability range of the low-complexity estimator, which gave me a clearer understanding of the method’s motivation and underlying mechanism. You also noted that the revised version will include standard deviations, significance tests, and cross-dataset estimation results to address the randomness issues associated with small datasets. Nevertheless, based on the current experimental scope and the level of substantiation in the paper, I will maintain my original rating.

---

> > > ### Author Response · Authors · 2025-12-03
> > > **Thank reviewer for their time and comments**
> > >
> > > Thank you very much for carefully reading our responses and for the thoughtful follow-up comment.
> > > We appreciate your clear summary and constructive suggestions, and we will incorporate the discussed revisions into the updated version.
> > > We are grateful for your evaluation and the time you devoted to our work.

---

### Meta-Review · Area_Chair_aZiH · 2025-12-07

**Summary:**

Reviewers pointed out concerns on model assumption,  computational complexity, performance inconsistency, design inconsistency, insufficient baseline, and the lack of an ablation study. Some of the concerns are partially answered. Two reviewers reply to the responses, and one of them increases the score. However, the remaining issues, especially from the reviewer gMeF, reveal this paper needs further polishing.

**Reviewer Concerns:**

- Reviewer gMeF pointed out some serious issues, including performance inconsistency, design inconsistency, insufficient baseline, and the lack of an ablation study. After the rebuttal, the design and performance inconsistencies are partially answered. However, the reviewer also pointed out further concerns about parameter analysis and performance inconsistency.
- Reviewer HRsQ’s concerns mainly focused on the alignment of the proposed fractal GCL and the core challenge of GCL, assumptions on fractal properties, the implementation details, and experimental settings. I think the responses to assumptions and settings are satisfactory, while some implementations are still confusing.
- Reviewer emw2 concerns the model assumptions and computation complexity. Although the clarification on the model assumption makes sense, the high computational complexity remains unsolved.

**Reviewer Scores:**

Reviewer gMeF has increased the score from 2 (reject) to 4 (weak reject) and pointed out further concerns.  Therefore, I think he may not further increase the scores.  Besides, reviewers’ concerns about assumptions are partially answered. Thus,  Reviewer HRsQ or emw2 may increase the score. Therefore, I think the final rating may be 6, 6, 4, 4.

---

### Decision · Program_Chairs · 2026-01-26

Reject